# Adaptive $Q$-Aid for Conditional Supervised Learning in Offline Reinforcement Learning

**Jeonghye Kim[1], Suyoung Lee[1], Woojun Kim[2], Youngchul Sung[1] ***
[1]KAIST  [2]Carnegie Mellon University

## Abstract

Offline reinforcement learning (RL) has progressed with return-conditioned supervised learning (RCSL), but its lack of stitching ability remains a limitation. We introduce $Q$-Aided Conditional Supervised Learning (QCS), which effectively combines the stability of RCSL with the stitching capability of $Q$-functions. By analyzing $Q$-function over-generalization, which impairs stable stitching, QCS adaptively integrates $Q$-aid into RCSL's loss function based on trajectory return. Empirical results show that QCS significantly outperforms RCSL and value-based methods, consistently achieving or exceeding the maximum trajectory returns across diverse offline RL benchmarks. The project page is available at https://beanie00.com/publications/qcs.

## 1 Introduction

Offline reinforcement learning (RL) is a vital framework for acquiring decision-making skills from fixed datasets, particularly when online interactions are impractical. This is especially relevant in fields such as robotics, autonomous driving, and healthcare, where the costs and risks of real-time experimentation are significant.

A promising approach in offline RL is return-conditioned supervised learning (RCSL) [10, 12, 20]. By framing offline RL as sequence modeling tasks, RCSL allows an agent to leverage past experiences and condition on the target outcome, facilitating the generation of future actions that are likely to achieve desired outcomes. This method builds on recent advancements in super-

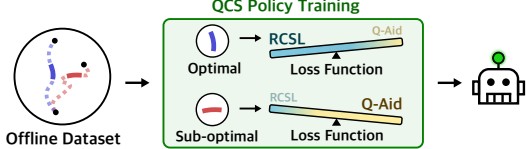

Figure 1: **Conceptual idea of QCS**: Follow RCSL when learning from optimal trajectories where it predicts actions confidently but the $Q$-function may stitch incorrectly. Conversely, refer to the $Q$-function when learning from sub-optimal trajectories where RCSL is less certain but the $Q$-function is likely accurate.

vised learning (SL) [41, 9, 11, 30], and thus benefits from the inherent stability and scalability of SL. However, RCSL is significantly limited by its lack of 'stitching ability', the ability to combine suboptimal trajectory segments to form better overall trajectories [13, 27, 53, 16, 7, 58]. As a result, its effectiveness is restricted to the best trajectories within the dataset.

Conversely, the $Q$-function possesses the ability to stitch together multiple sub-optimal trajectories, dissecting and reassembling them into an optimal trajectory through dynamic programming. Therefore, to address the weakness of RCSL, prior works have attempted to enhance stitching ability through the $Q$-function [53, 16]. However, these prior works employ the $Q$-function as a conditioning factor for RCSL and do not fully leverage the $Q$-function's stitching ability, resulting in either negligible performance improvements or even reduced performance. The primary challenge lies

---

*Correspondence to Youngchul Sung.

38th Conference on Neural Information Processing Systems (NeurIPS 2024).

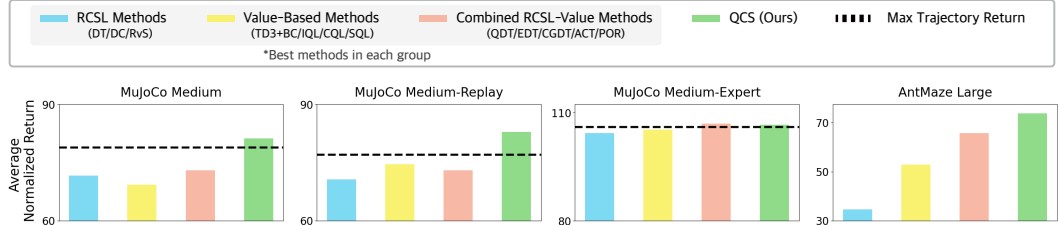

Figure 2: Mean normalized return in MuJoCo `medium`, `medium-replay`, `medium-expert`, and AntMaze `large`. The scores of RCSL, the value-based methods, and the combined methods represent the maximum mean performances within their respective groups. The full scores are in Section 6.2.

in the fact that utilizing the $Q$-function through conditioning, without managing the conditions for stable $Q$-guided stitching, can result in a sub-optimal algorithm.

In this work, we aim to fully synergize the stable and scalable learning framework of RCSL with the stitching ability of the $Q$-function. To effectively utilize the $Q$-function, it is crucial to identify when it can benefit RCSL and to integrate its assistance, termed $Q$-aid. Our contributions to achieving the effective utilization of $Q$-aid in RCSL are as follows: (1) We discovered that in-sample $Q$-learning on an expert dataset, which predominantly consists of optimal actions with similar $Q$-values within a constrained action range, causes the $Q$-function to receive improper learning signals and become over-generalized. (2) To prevent errors from this $Q$-generalization and to incorporate stitching ability within RCSL's stable framework, we propose $Q$-Aided Conditional Supervised Learning (QCS), which adaptively integrates $Q$-aid into the RCSL's loss function based on trajectory returns.

Despite its simplicity, the effectiveness of QCS is empirically demonstrated across offline RL benchmarks, showing significant advancements over existing state-of-the-art (SOTA) methods, including both RCSL and value-based methods. Especially, QCS surpasses the maximal dataset trajectory return across diverse MuJoCo datasets, under varying degrees of sub-optimality, as shown in Fig. 2. Furthermore, QCS significantly outperforms the baseline methods in the challenging AntMaze Large environment. This notable achievement underscores the practical effectiveness of QCS in offline RL.

## 2 Preliminaries

We consider a Markov Decision Process (MDP) [6], described as a tuple $\mathcal{M} = (\mathcal{S}, \mathcal{A}, \mathcal{P}, \rho_0, r, \gamma)$. $\mathcal{S}$ is the state space, and $\mathcal{A}$ is the action space. $\mathcal{P} : \mathcal{S} \times \mathcal{A} \mapsto \Delta(\mathcal{S})$ is the transition dynamics, $\rho_0 \in \Delta(\mathcal{S})$ is the initial state distribution, $r : \mathcal{S} \times \mathcal{A} \mapsto \mathbb{R}$ is the reward function, and $\gamma \in [0, 1)$ is a discount factor. The objective of offline RL is to learn a policy $\pi(\cdot|s)$ that maximizes the expected cumulative discounted reward, $\mathbb{E}_{a_t \sim \pi(\cdot|s_t), s_{t+1} \sim \mathcal{P}(\cdot|s_t, a_t)} \left[ \sum_{t=0}^{\infty} \gamma^t r(s_t, a_t) \right]$, using a static dataset $\mathcal{D} = \{\tau^{(i)}\}_{i=1}^{D}$ comprising a set of trajectories $\tau^{(i)}$. Each trajectory $\tau^{(i)}$ consists of transitions over a time horizon $T$, collected from an unknown behavior policy $\beta$.

### 2.1 Value-Based Offline Reinforcement Learning

Offline RL effectively employs off-policy RL techniques, allowing a divergence between the behavior policy $\beta$ used for data acquisition and the target policy $\pi$ being optimized [25, 14, 23]. Off-policy methods primarily utilize the $Q$-function, which is learned through temporal-difference (TD) bootstrapping. In actor-critic off-policy approaches, both the $Q$-function $\hat{Q}_\theta$ and the policy $\hat{\pi}$ are updated iteratively. This process can cause a shift in the action distribution, leading $\hat{\pi}$ to select actions that significantly deviate from those in the training dataset. These deviations can inadvertently result in overestimation errors, especially for out-of-distribution (OOD) actions, as offline RL cannot correct incorrect $Q$-values through interactions with the environment.

Unlike actor-critic methods, in-sample learning methods use only in-sample actions to learn the optimal $Q$-function, thereby preventing the querying of OOD action $Q$-values during training [34, 32, 23, 51]. Implicit $Q$-Learning (IQL) [23] is a representative in-sample learning method. It utilizes expectile regression, defined as $L_\eta^2(u) = |\eta - \mathbb{1}(u < 0)|u^2$ where $\eta \in [0.5, 1)$, to formulate the

asymmetrical loss function for the value network $V_\psi$. Through this loss, $V_\psi$ can approximate the implicit maximum of the TD target, $\max_a Q_{\hat\theta}(s, a)$. Formally, for a parameterized critic $Q_\theta(s, a)$ with a target critic $Q_{\hat\theta}(s, a)$, the value loss function is given by

$$\mathcal{L}_V(\psi) = \mathop{\mathbb{E}}_{(s,a)\sim\mathcal{D}} \left[ L_\eta^2 \left( Q_{\hat\theta}(s, a) - V_\psi(s) \right) \right]. \tag{1}$$

Intuitively, this loss function suggests placing more emphasis when $Q_{\hat\theta}$ is greater than $V_\psi(s)$. Subsequently, the critic network $Q_\theta$ is updated by treating the learned $V_\psi(s')$ as $\max_{a'\in\mathcal{D}(s')} Q_{\hat\theta}(s', a')$, where $\mathcal{D}(s')$ denotes the in-sample actions for the given state $s'$, i.e., $(s', a') \in \mathcal{D}$:

$$\mathcal{L}_Q(\theta) = \mathop{\mathbb{E}}_{(s,a,s')\sim\mathcal{D}} \left[ \left( r(s, a) + \gamma V_\psi(s') - Q_\theta(s, a) \right)^2 \right]. \tag{2}$$

We use IQL to pretrain the $Q$-function used to aid RCSL, as we found that this method, without conservatism during $Q$-function training, can provide good stitching ability when well integrated. A comparison with a different $Q$-learning method, CQL [25], is provided in Appendix H.1.

## 2.2 Return-Conditioned Supervised Learning (RCSL)

RCSL is an emerging approach to addressing challenges in offline RL. It focuses on learning the action distribution conditioned on *return-to-go* (RTG), defined as the cumulative sum of future rewards $\hat{R}_t = \sum_{t'=t}^T r_{t'}$ through supervised learning (SL) [10, 12, 20]. Due to the stability of SL, RCSL is capable of learning decision-making by extracting and mimicking useful information from the dataset. In particular, Decision Transformer (DT) [10] applies the Transformer architecture [41] to reframe the RL as a sequence modeling problem. It constructs input sequences to the Transformer by using sub-trajectories, each spanning $K$ timesteps and comprising RTGs, states, and actions: $\tau_{t-K+1:t} = (\hat{R}_{t-K+1}, s_{t-K+1}, a_{t-K+1}, ..., \hat{R}_{t-1}, s_{t-1}, a_{t-1}, \hat{R}_t, s_t)$. The model is then trained to predict the action $a_t$ based on $\tau_{t-K+1:t}$. Recently, Kim et al. [20] proposed Decision ConvFormer (DC) to simplify the attention module of DT and better model the local dependency in the dataset, yielding performance gains over DT with reduced complexity. These methods have shown effective planning capabilities, but they lack stitching ability, which causes difficulties with datasets that contain many sub-optimal trajectories. This will be discussed in more detail in Section 3.1.

## 2.3 Neural Tangent Kernel of $Q$-Function

The Neural Tangent Kernel (NTK) [19] provides insightful analysis of function approximation errors of $Q$-function, $Q_\theta$, especially those related to generalization. The NTK, denoted as $k_\theta(\bar{s}, \bar{a}, s, a)$, is defined as the inner product of two gradient vectors, $\nabla_\theta Q_\theta(\bar{s}, \bar{a})$ and $\nabla_\theta Q_\theta(s, a)$, i.e., $k_\theta(\bar{s}, \bar{a}, s, a) := \nabla_\theta Q_\theta(\bar{s}, \bar{a})^\top \nabla_\theta Q_\theta(s, a)$. The NTK offers a valuable perspective on the impact of parameter updates in function approximation, particularly in gradient descent scenarios. It essentially measures the degree of influence a parameter update for one state-action pair $(s, a)$ exerts on another pair $(\bar{s}, \bar{a})$. A high value of $k_\theta(\bar{s}, \bar{a}, s, a)$ implies that a single update in the $Q_\theta$ for the pair $(s, a)$ could lead to substantial changes for the pair $(\bar{s}, \bar{a})$. We guide the readers to Appendix D for a deeper understanding of the NTK.

## 3 When Is $Q$-Aid Beneficial for RCSL?

When is it beneficial for RCSL to receive assistance from the $Q$-function, denoted as $Q_\theta$, and how should this assistance be provided? To explore this, we trained two policies, RCSL policy and a max-$Q$ policy that selects the best action according to $Q_\theta$, on two different quality datasets from D4RL [13] MuJoCo, comparing their performances in Table 1. Note that the performance is not directly linked to the policy's accuracy across all states; even if the agent accurately predicts actions in several states, errors in some states can lead to path deviations and accumulated errors, resulting in a test-time distribution shift and a lower trajectory return. However, these results can provide insight into when $Q$-aid might be helpful.

For the RCSL algorithm, we used the Decision Transformer (DT) [10]. To train the max-$Q$ policy, we first trained the $Q_\theta$ using the in-sample $Q$-learning method outlined in Eqs. (1) and (2). Then, we extracted the max-$Q$ policy to select the action that directly maximizes $Q_\theta(s, \cdot)$ for each state $s$ by using a 3-layer MLP and the loss function $\mathcal{L}_{\text{max-}Q}(\phi) = \mathbb{E}_{s\sim\mathcal{D}} \left[ -Q_\theta\left( s, \text{max-}Q_\phi(s) \right) \right]$.

Table 1: Performance comparison of DT and max-$Q$ on `expert` and `medium-replay` quality datasets in MuJoCo.

| | halfcheetah-e | halfcheetah-m-r | hopper-e | hopper-m-r | walker2d-e | walker2d-m-r |
|---|---|---|---|---|---|---|
| DT | 91.4 ± 1.7 | 36.6 ± 0.8 | 110.1 ± 0.9 | 82.7 ± 7.0 | 109.2 ± 1.5 | 66.6 ± 3.0 |
| max-$Q$ | -4.1 ± 1.1 | 52.8 ± 0.4 | 1.8 ± 1.0 | 92.1 ± 2.6 | -0.2 ± 0.6 | 91.2 ± 1.9 |

Observing Table 1, we see that the dataset quality favoring RCSL contrasts with that benefiting the max-$Q$ policy. RCSL tends to perform well by mimicking actions in high-return trajectory datasets [29, 31]. However, this method is less effective with datasets predominantly containing suboptimal trajectories, even though RTG conditioning helps predict actions that yield higher returns. On the other hand, the max-$Q$ policy excels with suboptimal datasets but shows notably poor results with optimal datasets. From these observations, a motivating question arises: *"Why does the simple max-$Q$ policy outperforms RCSL on suboptimal datasets yet fails on optimal datasets? If so, how can we effectively combine the two methods to achieve optimal performance?"*

### 3.1 How Can Max-$Q$ Policy Surpass RCSL in Suboptimal Datasets?

We present a toy example demonstrating the limitation of RCSL, as illustrated in Fig. 3. Suppose the dataset is composed of two sub-optimal trajectories. At the initial state $s^1$, the agent has two options: the $\uparrow$ action connected to trajectory 2 (the orange trajectory) with an RTG of 5, and the $\rightarrow$ action connected to trajectory 1 (the purple trajectory) with an RTG of 6. RCSL makes the agent choose the $\rightarrow$ action with a high RTG and follow the path of trajectory 1, which is not optimal. This example demonstrates that RCSL alone is insufficient for the agent to learn to assemble the parts of beneficial sub-trajectories.

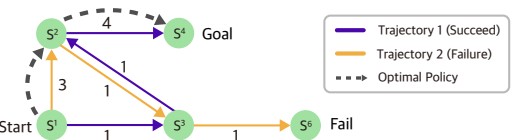

Figure 3: An example demonstrating the limit of RCSL: The dataset consists of two trajectories, with a time limit of $T = 3$ and a discount factor $\gamma = 1$. The black dashed arrow represents the optimal policy yielding a maximum return of 7.

In contrast, a $Q$-function can develop stitching ability. Consider the example in Fig. 3 again. We can compute the $Q$-values for the actions $\uparrow$ and $\rightarrow$ at state $s_1$ with dynamic programming: $Q(s^1, \uparrow) = 3 + \max\left(Q(s^2, \rightarrow), Q(s^2, \searrow)\right) = 7$, $Q(s^1, \rightarrow) = 1 + \max\left(Q(s^3, \rightarrow), Q(s^3, \nwarrow)\right) = 6$.

With the $Q$-values, the agent will select the $\uparrow$ action at $s^1$ and then the $\rightarrow$ action at $s^2$. Consequently, using the $Q$-function, the agent can select the optimal action that yields the maximum return of 7. Therefore, integrating RCSL with $Q$-function in situations with abundant sub-optimal trajectories can be beneficial for developing the stitching ability required for optimal decision-making.

### 3.2 Why Does Max-$Q$ Policy Struggle with Optimal Datasets?

Despite the potential advantages of using $Q$-values, incorporating values from a learned $Q$-function, $Q_\theta$, to aid RCSL can introduce errors due to inaccuracies in learning. These inaccuracies are particularly significant when $Q_\theta$ is learned from optimal trajectories. Suppose we have an optimal policy $\pi^*$. Optimal trajectories are visit logs containing actions performed by $\pi^*$, yielding the best $Q$-value for a given state $s$. Due to the stochasticity of $\pi^*$, multiple similar actions can be sampled from $\pi^*$, namely $a_1^*, a_2^*, \ldots, a_{n(s)}^* \sim \pi^*(\cdot|s)$ for a given state $s$. In this case, we have $Q^*(s, a_i^*) \approx Q^*(s, a_j^*)$ and $a_i^* \approx a_j^*$ $\forall i, j \in \{1, 2, \ldots, n(s)\}$ due to the optimality of these actions. When learning $Q_\theta$ from such limited information, where the values at the narrow action points are almost identical for each given state, it is observed that the learned $Q_\theta(s, a)$ tends to be over-generalized to the OOD action region. This means that the nearly identical value at the in-sample actions $a_1^*, a_2^*, \ldots, a_{n(s)}^*$ is extrapolated to OOD actions, yielding a nearly flat $Q$-value over the entire action space for each given state, i.e., $Q_\theta(s, a_{\text{OOD}}) \approx Q_\theta(s, a_1^*)$ with $Q_\theta(s, a)$ being a function of $s$ only. This over-generalization makes $Q_\theta$ noise-sensitive, potentially assigning high values to incorrect actions and causing state distribution shifts in the test phase, as shown in Fig. 7.

We present a simple experiment to verify that learning $Q_\theta$ indeed induces over-generalization when trained on optimal trajectories. The experiment consists of an MDP with one-dimensional discrete

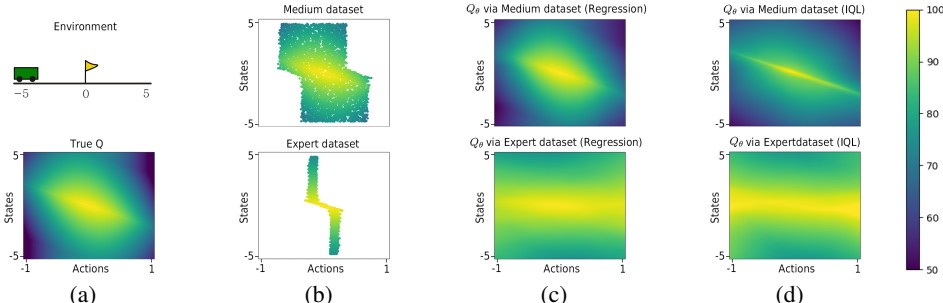

(a)  (b)  (c)  (d)

Figure 4: (a) the view of the environment and true $Q$ calculated through value iteration, (b) training datasets with color representing the true $Q$ for each sample, (c) $Q_\theta$ learned through regression with a medium dataset (upper) and an expert dataset (bottom), (d) $Q_\theta$ learned through IQL with a medium dataset (upper) and an expert dataset (bottom).

states and actions, each divided into 500 bins. This environment simulates a car, where the state indicates the agent's position, ranging from -5 to 5, as illustrated in Fig. 4 (a). The action range is between -1 and 1, allowing the agent to move according to the direction and twice the magnitude of the action. The objective is to reach position 0, which grants a reward of 100, while larger actions incur penalties given as $-30 \cdot a^2$. Due to its discrete nature, we can compute the true optimal $Q$-values through value iteration [37], which is shown in the bottom row of Fig. 4 (a).

With this environment, we generated two datasets, `medium` and `expert`. The `medium` dataset consisted of actions varying within the range of ±0.5 perturbed from the optimal action determined by the true optimal $Q$-values, while the `expert` dataset consisted of actions varying within the range of ±0.05 perturbed from the optimal action. (Refer to Fig. 4 (b)) We then adopted a 3-layer MLP as the structure of $Q_\theta$ and performed regression to follow the true $Q$-value at each sample point $(s, a)$ in the trajectories. Note that in-sample $Q$-learning can essentially be regarded as regression with the target value obtained from bootstrapping.

The learned $Q_\theta$ with the `medium` and `expert` datasets are shown in Fig. 4 (c). Indeed, the learned $Q_\theta$ with the `expert` dataset, containing nearly-optimal actions, shows that the value is flat over the entire action space for each state. This means that the nearly identical value of in-sample expert actions with a small spread is projected to the entire action space for each state. In contrast, the learned $Q_\theta$ with the `medium` dataset well estimates the true $Q$-function. This is because the medium dataset has diverse actions with diverse values for each state that facilitate the regression process. We additionally present the results from $Q_\theta$ obtained through IQL in Fig. 4 (d), which shows a similar trend to the results from regression.

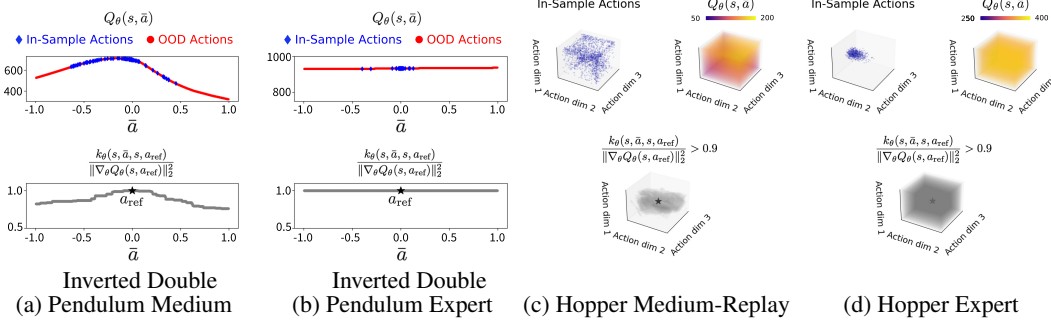

(a) Pendulum Medium    (b) Pendulum Expert    (c) Hopper Medium-Replay    (d) Hopper Expert

Figure 5: We present the estimated $Q_\theta(s, \bar{a})$ for $\bar{a} \in \mathcal{A}$ and the normalized NTK $k_\theta(s, \bar{a}, s, a_{\text{ref}})/\|\nabla_\theta Q_\theta(s, a_{\text{ref}})\|_2^2$ across four datasets with a 1D action space for Inverted Double Pendulum and a 3D action space for Hopper. In these figures, we fix the state $s$ and the fixed reference action $a_{\text{ref}}$ at zero (marked as ⋆), and sweep over all actions $\bar{a} \in \mathcal{A}$. For Hopper, we use axes for action dimensions and color to represent $Q$-values in 3D plots. Additionally, in the NTK plot, we only include the high-NTK regions for values over 0.9. Refer to Appendix E for details.

The over-generalization tendency in $Q_\theta$ with optimal trajectories is not limited to the simple experiment above but also applies to complex RL tasks. We analyze how $Q_\theta(s, \cdot)$ varies over the action space in the Gym Inverted Double Pendulum [8] and MuJoCo Hopper environments [39, 8] trained on `expert` and `medium`-quality datasets with IQL. The details of the analysis are in Appendix E. As depicted in the upper row of Fig. 5 (a) and (b), and on the left side of the upper row of (c) and (d), the `expert` dataset shows concentrated action distribution, while the `medium` dataset has a broader spread, as expected. The concentration and similarity of true $Q$-values of the actions for a given state in the `expert` dataset cause over-generalization in $Q_\theta(s, \cdot)$, yielding constant flat values across the entire action space. This is further supported by the results in Appendix E.2, which visualize the weights of the learned $Q$-function.

For a deeper understanding of the over-generalization in $Q_\theta$, we analyze the gradient similarity, captured as the Neural Tangent Kernel (NTK), between an arbitrary action $\bar{a}$ and the reference action $a_{\text{ref}}$ for a given state $s$. A higher NTK value signifies that the gradient update of $Q_\theta(s, a_{\text{ref}})$ has a substantial impact on $Q_\theta(s, \bar{a})$. This indicates that even when $a_{\text{ref}}$ and $\bar{a}$ are dissimilar or unrelated actions, a high NTK value suggests that the $Q_\theta$ network is misjudging the relationship between these actions and over-generalizing. In Fig. 5, $Q_\theta$ trained with the `expert` dataset shows uniformly high normalized NTK values across actions, indicating that the gradient at one action equally affects all others. In contrast, $Q_\theta$ trained with the `medium` dataset shows NTK values that are higher near the reference action and decrease with action distance, reflecting more precise generalization. This analysis reveals that datasets consisting of optimal trajectories exhibit more aggressive generalization, which can negatively impact the accuracy of the learned $Q$-function in offline RL.

## 4  $Q$-Aided Conditional Supervised Learning

According to Section 3, RCSL faces challenges with suboptimal datasets, whereas $Q_\theta$ can effectively serve as a critic for stitching ability, favoring the use of $Q$-aid. In contrast, in an optimal dataset, $Q_\theta$ tends to over-generalize, leading to inaccuracies of learned $Q_\theta$, while RCSL excels by mimicking the optimal behavior without requiring external assistance. Recognizing this dynamic, it is crucial to integrate $Q$-aid into RCSL adaptively. The following subsections explain how to effectively adjust the level of $Q$-aid and facilitate the integration of the two methods, leading to the proposal of $Q$-Aided Conditional Supervised Learning (QCS).

### 4.1  Controlling $Q$-Aid Based on Trajectory Returns

Given the complementary relationship, how can we adjust the degree of $Q$-aid? Since RCSL's preference for mimicking datasets and the $Q$-function's over-generalization issue is tied to trajectory optimality, we can apply varying degrees of $Q$-aid based on the trajectory return for each sub-trajectory in RCSL. Therefore, we set the degree of $Q$-aid, denoted as the QCS weight $w(R(\tau))$ for a trajectory $\tau$, as a continuous, monotone-decreasing function of the return of $\tau$, $R(\tau)$, such that

$$\forall \tau_1, \tau_2, \ \ R(\tau_1) < R(\tau_2) \Rightarrow w(R(\tau_1)) \geq w(R(\tau_2)),$$

where continuity is imposed for gradual impact change. Among various choices, we find that simple options such as linear decay are sufficient to produce good results, i.e., $w(R(\tau)) = \lambda \cdot (R^* - R(\tau))$ with some $\lambda > 0$, where $R^*$ represents the optimal return of the task. Practically, $R^*$ can be obtained from an expert dataset or from the maximum value in the dataset. For details on how to calculate $R^*$, please refer to Appendix F. Note that $R(\tau)$ differs from RTG $\hat{R}_t$ which is the sum of future rewards after timestep $t$ and decreases as timestep $t$ goes, thereby failing to represent the trajectory's optimality accurately.

### 4.2  Integrating $Q$-Aid into the RCSL Loss Function

Instead of using the $Q$-function as the conditioning factor for RCSL as in previous works, we propose a more explicit approach by integrating $Q$-assistance into the loss function and dynamically adjusting the degree of assistance based on Section 4.1. As a result, the overall policy loss is given as follows:

$$\mathcal{L}_\pi^{\text{QCS}}(\phi) = \mathbb{E}_{\tau \sim \mathcal{D}} \left[ \frac{1}{K} \sum_{j=0}^{K-1} \underbrace{\| a_{t+j} - \pi_\phi(\tau_{t:t+j}) \|_2^2}_{\text{RCSL}} - \underbrace{\lambda \cdot (R^* - R(\tau))}_{\text{QCS weight}} \cdot \underbrace{Q_\theta^{\text{IQL}}(s_{t+j}, \pi_\phi(\tau_{t:t+j}))}_{Q \text{ Aid}} \right], \quad (3)$$

where $Q_\theta^{\mathrm{IQL}}(\cdot, \cdot)$ denotes the fixed $Q$-function pretrained with IQL. $R(\tau)$ is the return of the entire trajectory $\tau$ containing the sub-trajectory $\tau_{t:t+K-1}$. The overall input to the policy at time $t$ is the sub-trajectory of context length $K$ starting from time $t$, $\tau_{t:t+K-1} = \left(\hat{R}_t, s_t, a_t, \ldots, \hat{R}_{t+K-1}, s_{t+K-1}\right) \subset \tau$.

Our new loss function enables adaptive learning strategies depending on the trajectory's quality to which the subtrajectory belongs. For optimal trajectories, action selection follows RCSL. On the other hand, for suboptimal trajectories $\tau$ with $R(\tau) < R^*$, the $Q$-aid term kicks in and its impact increases as $R(\tau)$ decreases. We describe the details of the QCS weight $w(R(\tau))$ and the policy update with the loss function in Appendix J.2 and our full algorithm's pseudocode in Appendix A.

### 4.3 Implementation

**Base Architecture.** For implementing $\pi_\phi$, a general RCSL policy can be used. When $K = 1$, meaning only the current time step is considered to estimate the action, we use an MLP network. When $K \geq 2$, we use a history-based policy network, such as DT [10] or DC [20].

**Conditioning.** We consider two conditioning approaches as proposed by RvS [12]: one for tasks maximizing returns and the other for tasks aiming at reaching specific goals. For return-maximizing tasks, we employ RTG conditioning, and our algorithm is named QCS-R. For goal-reaching tasks, we additionally use subgoal conditioning, and our algorithm is named QCS-G. For subgoal selection, we randomly select a state that the agent will visit in the future. The ablations on conditioning are in Appendix H.2.

## 5 Related Work

**Prompting RCSL with Dynamic Programming.** Recent studies have recognized the limitations of RCSL in stitching abilities [27, 7, 58]. Our work contributes to the ongoing efforts to imbue RCSL with this capability. Notably, $Q$-learning Decision Transformer (QDT) [53] and Advantage Conditioned Transformer (ACT) [16] have proposed integrating dynamic programming into RCSL by modifying the RTG prompt to $Q$-value or advantage prompt. Our approach, QCS, parallels these efforts by leveraging dynamic programming for action guidance and trajectory stitching. However, unlike these methods, which implicitly incorporate dynamic programming through conditioning, QCS explicitly augments its loss function with the learned $Q$-function.

**Incorporating RCSL with Stitching Ability.** In a distinct vein, recently proposed Critic-Guided Decision Transformer (CGDT) [43] identifies the gap between target RTG and expected returns of actions as key to RCSL's limited stitching. To mitigate this, it adjusts DT's output with the critic network's Monte-Carlo return predictions and target RTG. In contrast, QCS uses $Q$-values learned through dynamic programming to guide actions, enhancing stitching ability explicitly. Another approach, the Elastic Decision Transformer (EDT) [49], recommends variable context lengths during inference, using longer contexts for optimal trajectories and shorter ones for sub-optimal trajectories to identify optimal paths better. QCS similarly adapts based on trajectory optimality but differentiates itself by modifying its learning approach during training, leveraging the complementary strengths of the $Q$-function and RCSL.

Furthermore, POR [50] integrates imitation learning techniques with stitching ability by generating high-value states using additional networks and value functions. These states are then used as conditions for predicting actions. Unlike QCS, which focuses on action stitching, POR emphasizes state stitching, allowing agents to choose actions that lead to high-value states, albeit with the need for additional networks. By concentrating on action stitching, QCS can avoid the computational demands associated with high-dimensional state prediction.

**State-Adaptive Balance Coefficient** Regarding the sub-trajectory-adaptive weight used in QCS, FamO2O [42] employs state-adaptive weight coefficients to balance policy improvement and constraints in the offline-to-online RL framework. Although FamO2O is an offline-to-online method that incorporates additional online samples, we provide a performance comparison with this work in Appendix G.2 to further demonstrate the effectiveness of QCS.

# 6 Experiments

In the experiment section, we conduct various experiments across different RL benchmarks to answer the following questions:

- How well does QCS perform in decision-making compared to prior SOTA methods across datasets of varying quality and tasks with diverse characteristics, especially those requiring stitching ability?
- To what extent does the dynamic nature of QCS weights, informed by trajectory return, contribute to effective decision-making, and how robust are these dynamic weights to hyperparameters?
- Can QCS effectively acquire stitching ability while preventing test-time distribution shift?

## 6.1 Experimental Setup

**Baseline Methods.** To address a range of questions, we conduct a comprehensive benchmarking against 12 representative baselines that are state-of-the-art in each category. For the value-based category, we assess 4 methods: TD3+BC [14], IQL [23], and CQL [25], SQL [52]. For RCSL, we assess 3 methods: DT [10], DC [20], RvS [12]. Additionally, we evaluate 5 advanced RCSL methods proposed to integrate stitching capabilities: QDT [53], EDT [49], CGDT [43], ACT [16], and POR [50]. For more details on the setup and the baselines, refer to Appendix B.

**Benchmarks.** We evaluated QCS against various baselines using datasets with diverse characteristics, including tasks focused on return maximization or goal-reaching, and those with dense or sparse rewards and varying sub-optimality levels.

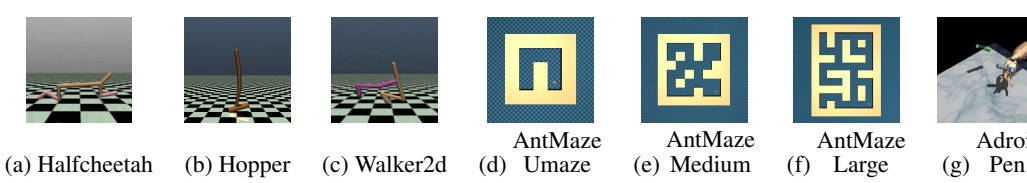

| (a) Halfcheetah | (b) Hopper | (c) Walker2d | (d) AntMaze Umaze | (e) AntMaze Medium | (f) AntMaze Large | (g) Adroit Pen |

Figure 6: Views of tasks used in our experiments.

Our primary focus was on the D4RL [13] MuJoCo, AntMaze, and Adroit domains. The MuJoCo domain [39, 8] features several continuous locomotion tasks with dense rewards. We conducted experiments in three environments: Halfcheetah, Hopper, and Walker2d, utilizing three distinct v2 datasets—`medium`, `medium-replay`, and `medium-expert`—each representing different levels of data quality. AntMaze is a domain featuring goal-reaching environments with sparse rewards, encompassing variously sized and shaped maps. It is an ideal testing bed for evaluating an agent's capability to stitch trajectories and perform long-range planning. We conduct experiments using six v2 datasets: `umaze`, `umaze-diverse`, `medium-play`, `medium-diverse`, `large-play`, and `large-diverse`, where `umaze`, `medium`, and `large` indicate map sizes, and `play` and `diverse` refer to data collection strategies. The Adroit domain [35] comprises various tasks designed to evaluate the effectiveness of algorithms in high-dimensional robotic manipulation tasks. In our experiments, we utilize the `human` and `cloned` datasets for the pen task.

Performance results for the MuJoCo and AntMaze domains are presented in Table 2 and Table 3, while results for the Adroit domain are included in Appendix G.

**Hyperparameters and Backbone Architecture.** We adopted two sets of hyperparameters per domain to determine the gradient of the monotonic decreasing function $w(R(\tau))$. The detailed hyperparameters we used are provided in Appendix J, and the impact of $\lambda$ is detailed in Appendix 6.3. Additionally, we implemented QCS based on DT, DC, and a simple MLP, and compared the performance of each. Detailed results for each architectural choice are provided in Appendix H.2. We observed that the DC-based approach performs best, although the performance gap is minor.

**Evaluation Metric.** In all evaluations of QCS, we assess the expert-normalized returns [13] of 10 episodes at each evaluation checkpoint (every $10^3$ gradient steps). Subsequently, we compute the running average of these normalized returns over ten consecutive checkpoints. We report the mean and standard deviations of the final scores across five random seeds.

## 6.2 Overall Performance

Table 2: Performance of QCS and baselines in the MuJoCo domain. The dataset names are abbreviated as follows: `medium` to 'm', `medium-replay` to 'm-r', `medium-expert` to 'm-e'. The boldface numbers denote the maximum score or comparable one among the algorithms.

| Dataset | Value-Based Method | | | | RCSL | | | Combined Method | | | | | Ours |
|---|---|---|---|---|---|---|---|---|---|---|---|---|---|
| | TD3+BC | IQL | CQL | SQL | DT | DC | RvS-R | QDT | EDT | CGDT | ACT | POR | QCS-R |
| halfcheetah-m | 48.3 | 47.4 | 44.0 | 48.3 | 42.6 | 43.0 | 41.6 | 42.3 | 42.5 | 43.0 | 49.1 | 48.8 | **59.0** ± 0.4 |
| hopper-m | 59.3 | 66.3 | 58.5 | 75.5 | 67.6 | 92.5 | 60.2 | 66.5 | 63.5 | **96.9** | 67.8 | 78.6 | **96.4** ± 3.7 |
| walker2d-m | 83.7 | 78.3 | 72.5 | 84.2 | 74.0 | 79.2 | 71.7 | 67.1 | 72.8 | 79.1 | 80.9 | 81.1 | **88.2** ± 1.1 |
| halfcheetah-m-r | 44.6 | 44.2 | 45.5 | 44.8 | 36.6 | 41.3 | 38.0 | 35.6 | 37.8 | 40.4 | 43.0 | 43.5 | **54.1** ± 0.8 |
| hopper-m-r | 60.9 | 94.7 | 95.0 | 99.7 | 82.7 | 94.2 | 73.5 | 52.1 | 89.0 | 93.4 | 98.4 | 98.9 | **100.4** ± 1.1 |
| walker2d-m-r | 81.8 | 73.9 | 77.2 | 81.2 | 66.6 | 76.6 | 60.6 | 58.2 | 74.8 | 78.1 | 56.1 | 76.6 | **94.1** ± 2.0 |
| halfcheetah-m-e | 90.7 | 86.7 | 91.6 | 94.0 | 86.8 | 93.0 | 92.2 | - | - | 93.6 | **96.1** | 94.7 | 93.3 ± 1.8 |
| hopper-m-e | 98.0 | 91.5 | 105.4 | **111.8** | 107.6 | 110.4 | 101.7 | - | - | 107.6 | 111.5 | 90.0 | 110.2 ± 2.4 |
| walker2d-m-e | 110.1 | 109.6 | 108.8 | 110.0 | 108.1 | 109.6 | 106.0 | - | - | 109.3 | 113.3 | 109.1 | **116.6** ± 2.4 |
| average | 75.3 | 77.0 | 77.6 | 83.1 | 74.7 | 82.2 | 71.7 | - | - | 82.4 | 79.6 | 80.1 | **90.3** |

Table 3: Performance of QCS and baselines in the AntMaze domain. The dataset names are abbreviated as follows: `umaze` to 'u', `umaze-diverse` to 'u-d', `medium-play` to 'm-p', `medium-diverse` to 'm-d', `large-play` to 'l-p', and `large-diverse` to 'l-d'. The boldface numbers denote the maximum score or comparable one among the algorithms.

| Dataset | Value-Based Method | | | | RCSL | | | | Combined | Ours | |
|---|---|---|---|---|---|---|---|---|---|---|---|
| | TD3+BC | IQL | CQL | SQL | DT | DC | RvS-R | RvS-G | POR | QCS-R | QCS-G |
| antmaze-u | 78.6 | 87.5 | 74.0 | **92.2** | 65.6 | 85.0 | 64.4 | 65.4 | 90.6 | **92.7** ± 3.9 | **92.5** ± 4.6 |
| antmaze-u-d | 71.4 | 62.2 | **84.0** | 74.0 | 51.2 | 78.5 | 70.1 | 60.9 | 71.3 | 72.3 ± 12.4 | 82.5 ± 8.2 |
| antmaze-m-p | 10.6 | 71.2 | 61.2 | 80.2 | 4.3 | 33.2 | 4.5 | 58.1 | **84.6** | 81.6 ± 6.9 | **84.8** ± 11.5 |
| antmaze-m-d | 3.0 | 70.0 | 53.7 | **79.1** | 1.2 | 27.5 | 7.7 | 67.3 | **79.2** | 79.5 ± 5.8 | 75.2 ± 11.9 |
| antmaze-l-p | 0.2 | 39.6 | 15.8 | 53.2 | 0.0 | 4.8 | 3.5 | 32.4 | 58.0 | 68.7 ± 7.8 | **70.0** ± 9.6 |
| antmaze-l-d | 0.0 | 47.5 | 14.9 | 52.3 | 0.5 | 12.3 | 3.7 | 36.9 | 73.4 | 70.6 ± 5.6 | **77.3** ± 11.2 |
| average | 27.3 | 63.0 | 50.6 | 71.8 | 20.5 | 40.2 | 25.6 | 53.5 | 76.2 | 77.6 | **80.4** |

As shown in Table 2 and Table 3, QCS significantly outperforms prior value-based methods, RCSL, and combined methods across the datasets. Specifically, QCS outperforms both IQL and DC, upon which it is based, across all datasets, unlike the contradictory results between RCSL and the max-$Q$ policy shown in Table 1. This empirically confirms that QCS successfully combines the strengths of both RCSL and the $Q$-function. A particularly remarkable achievement of QCS is its ability to substantially improve efficiency in goal-reaching tasks, AntMaze, especially in Large environments, where prior RCSL methods exhibited notably low performance. This enhancement is largely attributed to the stitching ability introduced by the $Q$-aid of QCS. These results underscore QCS's robustness and superiority in a wide array of offline RL contexts. The training curves for Tables 2 and 3 are shown in Appendix I, demonstrating stable learning curves across all datasets.

## 6.3 Ablation Studies

To further analyze how each design element influences performance, we conducted additional experiments. More ablation studies are detailed in Appendix H, including the use of $Q$-function trained by CQL and the impact of base architecture and conditioning.

**The Importance of Weights Relative to Trajectory Return.**

To assess the impact of dynamically setting the QCS weight $w(R(\tau))$ based on trajectory return, we compare our approach with a constant QCS weight, $w(R(\tau)) = c$. We test five constant weights $c \in \{1, 2.5, 5, 7.5, 10\}$ and report the maximum score among these values in Table 4. The QCS method with the dynamic weight based on trajectory return outperforms the highest scores obtained with various constant weight settings across datasets, as shown in Table 4.

Table 4: Comparison of constant QCS weight and the dynamic weight.

| Dataset | Constant Weight | Dynamic Weight |
|---|---|---|
| mujoco-m | 74.7 | **81.2** |
| mujoco-m-r | 75.4 | **82.9** |
| mujoco-m-e | 104.2 | **106.7** |

This demonstrates that our dynamic weight control, grounded in trajectory return, is more effective and robust in integrating $Q$-aids.

**Impact of the QCS weight $\lambda$.** We examined the effect of $\lambda$ by varying it from 0.2 to 1.5. As shown in Table 5, except for the walker2d-medium, we found that even the smallest values achieved with changing $\lambda$ either matched or surpassed the performance of existing value-based methods and RCSL's representative methods, including IQL, CQL, DT, DC, and RvS. This demonstrates QCS's relative robustness regarding hyperparameters. For walker2d-medium, we found that performance begins to decrease when $\lambda$ exceeds the initial setting of 0.5. While increasing the gradient steps from 500K to 1M improves performance at $\lambda = 1$, further increasing $\lambda$ to 1.5 leads to greater instability.

Table 5: Performance of QCS in the Mujoco domain with varying $\lambda$ values. The boldface numbers denote the maximum score or a comparable one.

|  | $\lambda = 0.2$ | $\lambda = 0.5$ | $\lambda = 1$ | $\lambda = 1.5$ |
|---|---|---|---|---|
| halfcheetah-medium | 53.7 ± 0.4 | 57.7 ± 0.3 | **59.0** ± 0.4 | **59.0** ± 0.2 |
| hopper-medium | 89.4 ± 5.6 | **96.4** ± 3.7 | 95.7 ± 3.5 | 88.8 ± 6.2 |
| walker2d-medium | 83.9 ± 4.7 | **88.2** ± 1.1 | 75.5±7.1 (500K) / **87.6**±3.9 (1M) | 60.7 ± 11.2 |
| halfcheetah-medium-replay | 52.0 ± 0.8 | 52.8 ± 0.5 | **54.1** ± 0.8 | **54.2** ± 0.6 |
| hopper-medium-replay | 98.5 ± 2.4 | **100.4** ± 1.1 | 99.4 ± 2.1 | **100.5** ± 0.7 |
| walker2d-medium-replay | 83.3 ± 5.7 | 93.2 ± 2.5 | **94.1** ± 2.0 | 92.3 ± 3.7 |

**Test Time State Distribution Shift.**

To validate whether QCS effectively acquires stitching ability while preventing a shift in the test-time state distribution, as discussed in Section 3.2, we present Fig. 7. This figure compares the state distributions explored by RCSL, max-$Q$, and QCS policies during evaluation. RCSL and max-$Q$, representing QCS's extremes, were trained using specific loss configurations: RCSL loss as QCS loss in Eq. 3 with $\lambda = 0$ and max-$Q$ loss as QCS loss without the RCSL term, i.e., selecting actions as $\mathrm{argmax}_{a \in \mathcal{A}} Q_\theta^{\mathrm{IQL}}(s, a)$. Fig. 7 illustrates RCSL's adherence to dataset states, contrasting with the notable state distribution shift of the max-$Q$ policy. QCS inherits RCSL's stability but surpasses its performance, indicating an effective blend of transition recombination without straying excessively from the state distribution.

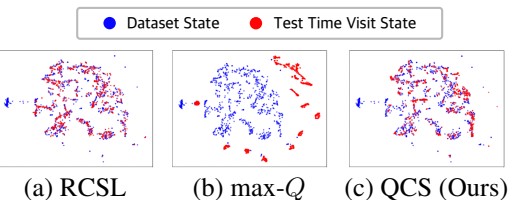

(a) RCSL     (b) max-$Q$     (c) QCS (Ours)

Figure 7: t-SNE [40] analysis of states visited by policies trained with RCSL, max-$Q$ ($\mathrm{argmax}_{a \in \mathcal{A}} Q_\theta^{\mathrm{IQL}}(s, a)$), and QCS losses during evaluation, alongside dataset's states in `walker2d-medium`.

## 7 Conclusion

In conclusion, QCS effectively combines the stability of RCSL with the stitching ability of the $Q$-function. Anchored by thorough observation of $Q$-function generalization error, QCS adeptly modulates the extent of $Q$-assistance. This strategic fusion enables QCS to exceed the performance of existing SOTA methods in both efficacy and stability, particularly in complex offline RL benchmarks encompassing a wide range of optimality.

In addressing our initial motivating question on integrating RCSL and $Q$-function, QCS opens up promising future research directions. While we have established a correlation between trajectory return and the mixing weight, we have considered simple linear weights to control the level of $Q$-aid. It is also plausible that the mixing weight might be influenced by other dataset characteristics, such as the dimensions of the state and actions. We believe QCS will stand as a motivating work, inspiring new advancements in the field.

**Acknowledgments**

This work was supported in part by Institute of Information & Communications Technology Planning & Evaluation (IITP) grant funded by the Korea government (MSIT) (No.2022-0-00469, Development of Core Technologies for Task-oriented Reinforcement Learning for Commercialization of Autonomous Drones, 50%) and in part by the National Research Foundation of Korea (NRF) grant funded by the Korea government (MSIT) (NRF-2021R1A2C2009143 Information Theory-Based Reinforcement Learning for Generalized Environments, 50%).

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

# Appendices

## A Pseudocode

The QCS algorithm first learns the $Q$-function using a dynamic programming and then trains the policy based on the aid of the learned $Q$-function. Detailed pseudocode can be found in Algorithm 1. In this work, we utilized IQL [23] as the $Q$ training algorithm, but other $Q$ training algorithms can be employed. A comparison with using a $Q$-function trained by CQL [25] can be found in Appendix H.1.

---

**Algorithm 1** IQL-aided QCS

---

    **Hyperparameters:** Total critic gradient steps $M$, critic learning rate $\alpha_{\text{critic}}$, target update rate $\chi$,
                        total policy gradient steps $N$, policy learning rate $\alpha_{\text{policy}}$, context length $K$
    **Initialize parameters:** $\theta$, $\hat{\theta}$, $\psi$, and $\phi$

    // IQL Pretraining
    **for** $m = 1$ **to** $M$ **do**
        $\psi \leftarrow \psi - \alpha_{\text{critic}} \nabla \mathcal{L}_V(\psi)$ using Eq. 1
        $\theta \leftarrow \theta - \alpha_{\text{critic}} \nabla \mathcal{L}_Q(\theta)$ using Eq. 2
        $\hat{\theta} \leftarrow \chi\theta + (1 - \chi)\hat{\theta}$
    **end for**

    // QCS Policy Training
    **for** $n = 1$ **to** $N$ **do**
        Sample trajectory $\tau \sim \mathcal{D}$
        Sample sub-trajectory $\tau_{t:t+K-1} \sim \tau$ with random initial timestep $t$
        $\phi \leftarrow \phi - \alpha_{\text{policy}} \nabla \mathcal{L}_\pi^{\text{QCS}}(\phi)$ using Eq. 3
    **end for**

---

## B Baseline Details

We evaluated the performance of QCS against twelve different baseline methods. This group consists of four value-based methods: TD3+BC [14], IQL [23], CQL [25] and SQL [52]; three RCSL algorithms: DT [10], DC [20], and RvS [12]; and five combined methods that signify progress in RCSL by integrating stitching abilities: QDT [53], EDT [49], and CGDT [43], ACT [16], and POR [50]. The performance for these baselines was sourced from their respective original publications, with two exceptions. For CQL [25], the performance data in the original paper was based on the MuJoCo v0 environment, which differs from the v2 version used in our study. Therefore, for CQL, we referenced the performance score reported in [23] to ensure a consistent and fair comparison across all methods.

In addition, for `antmaze-medium` and `antmaze-large`, since there were no reported DT [10] and DC [20] scores, we conducted evaluations using the official codebase. When training on `antmaze-medium` and `large`, we used 512 as the embedding dimension in the default hyperparameter setting. We found that removing the positional embedding slightly improved performance, as also discussed in Zheng et al. [57], so we trained without it. For the target RTG, we used values of 1 and 100 and reported the higher score obtained from the two values.

## C Dataset Return Distributions

To gain a deeper understanding of the scenarios in which offline RL is applied and the necessity of learning good policies, we plotted the trajectory return distributions for three different datasets in each of the three MuJoCo environments in Fig. 8. For these return distribution histograms, we set the number of bins to 50. The 'Count' label denotes the number of trajectories corresponding to each normalized return.

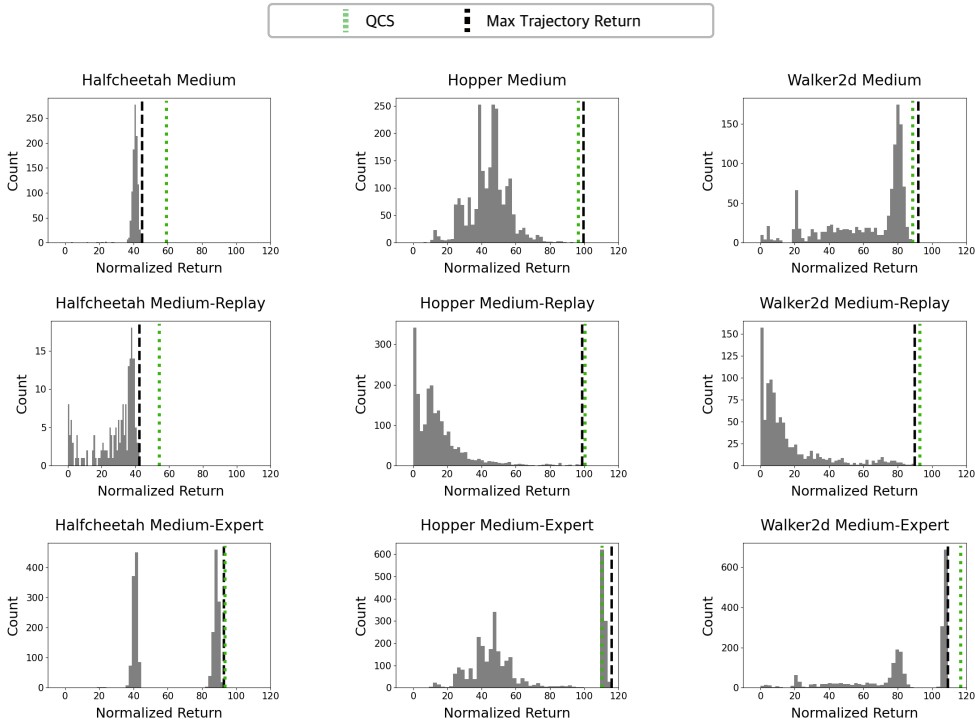

Figure 8: Distribution of trajectory returns in the MuJoCo datasets, including the dataset's maximum trajectory return and the QCS score.

As shown in Fig. 8, the `medium-replay` datasets encompass wide varieties of returns. Additionally, the `medium-expert` dataset, a combination of the medium and expert datasets, exhibits two peaks, indicating a division in the range of returns. This observation reveals that each dataset exhibits a distinct distribution pattern of returns. In this graph, alongside the return distribution, the dataset's maximum trajectory return and the score of our QCS method are also presented. QCS is observed to achieve results that are close to or surpass the maximum return. This is particularly notable in datasets with diverse return distribution characteristics, such as the `medium-replay` dataset, where the distribution of low return trajectories is prevalent.

## D  Brief Derivation of the Neural Tangent Kernel

To understand the influence of parameter updates in function approximation across different state-action pairs, the Neural Tangent Kernel (NTK) emerges as a crucial tool [19, 28, 1, 26, 56]. For comprehensive insights, we direct readers to Achiam et al. [1], while here we distill the essential concepts. The NTK framework becomes particularly relevant in the context of deep $Q$-learning, where the parameterized $Q$-function, denoted as $Q_\theta$, is updated as follows [37]:

$$\theta' = \theta + \alpha \mathop{\mathbb{E}}_{s,a\sim\rho} \left[ \delta_\theta(s,a) \nabla_\theta Q_\theta(s,a) \right], \tag{4}$$

where $\alpha$ is the learning rate, $\rho$ is the distribution of transitions in the dataset, and $\delta_\theta(s,a) = r(s,a) + \gamma \max_{a'} Q_\theta(s',a') - Q_\theta(s,a)$ is the temporal difference (TD) error. On the other hand, the first-order Taylor expansion around $\theta$ at an out-of-sample pair $(\bar{s}, \bar{a})$ yields

$$Q_{\theta'}(\bar{s},\bar{a}) = Q_\theta(\bar{s},\bar{a}) + \nabla_\theta Q_\theta(\bar{s},\bar{a})^\top (\theta' - \theta). \tag{5}$$

Substituting (4) into (5), we have

$$Q_{\theta'}(\bar{s},\bar{a}) = Q_\theta(\bar{s},\bar{a}) + \alpha \mathop{\mathbb{E}}_{s,a\sim\rho} \left[ k_\theta(\bar{s},\bar{a},s,a) \delta_\theta(s,a) \right], \tag{6}$$

where $k_\theta(\bar{s},\bar{a},s,a)$, referred to as the NTK, is defined as the inner product between two gradient vectors $\nabla_\theta Q_\theta(\bar{s},\bar{a})$ and $\nabla_\theta Q_\theta(s,a)$, i.e.,

$$k_\theta(\bar{s},\bar{a},s,a) := \nabla_\theta Q_\theta(\bar{s},\bar{a})^\top \nabla_\theta Q_\theta(s,a). \tag{7}$$

(6) together with (7) explains how the parameter update with function approximation for a sample pair $(s, a)$ affects the $Q$-value change for another sample pair $(\bar{s}, \bar{a})$. When the NTK $k_\theta(\bar{s}, \bar{a}, s, a)$ is large, the TD-error $\delta_\theta(s, a)$ has a more pronounced impact on the update difference $Q_{\theta'}(\bar{s}, \bar{a}) - Q_\theta(\bar{s}, \bar{a})$. Thus, the single update based on the TD-error at a sample pair $(s, a)$ can induce a significant change in the $Q$-function for another pair $(\bar{s}, \bar{a})$.

# E   Details and Extended Analysis of $Q$-Function and NTK

In Section 3.2, we conduct an NTK analysis of the $Q$-function trained with IQL in the Inverted Double Pendulum and Hopper environments, which have state dimensions of 2 and 11, and action dimensions of 1 and 3, respectively. This section details the analysis methods, provides extended results, and offers further clarity by presenting the action distributions for the datasets of each environment.

## E.1   Analysis Methods

**Inverted Double Pendulum.**     We chose the Inverted Double Pendulum for analysis of $Q$-values and NTK due to its one-dimensional action space. For training the $Q$-function, as no prior open-source offline dataset existed for this environment, we first created one. The dataset was generated by training an online Soft Actor-Critic [18], using an implementation in RLkit, available at `https://github.com/rail-berkeley/rlkit.git`.

We created two datasets: `expert` and `medium`. The `expert` dataset consists of $10^5$ samples generated by an optimal policy, while the `medium` dataset includes $10^5$ samples from a medium policy, whose performance is approximately one-third of the optimal policy. Given the continuous nature of the state and action spaces in the Inverted Double Pendulum, which complicates analysis, we initially quantized both spaces. For state quantization, we set the range from a minimum of -5 to a maximum of 10 (the minimum and maximum values across all dimensions in both datasets) and divided each dimension into 80 equal segments. For action quantization, the range was set from -1 to 1, divided into 500 equal segments.

When plotting the $Q$-values, we calculated the $Q$-value for each quantized state across all quantized actions. Fig. 5 shows the results for the state chosen in each dataset, based on the highest count of in-sample actions. In the NTK analysis, we computed the following equation for the reference action and the remaining quantized actions with index $i \in 1, \ldots, 500$, where $a_1 = -1$, $a_{500} = 1$, and $a_{\text{ref}} = 0$.

**MuJoCo Hopper.**     For Hopper environment, we use open-source D4RL [13] `hopper-expert` and `hopper-medium-replay` datasets. In the case of Hopper, similar to what was done in the Inverted Double Pendulum environment, we quantized the continuous state and action space for analysis. More specifically, for the state space, we divided the values of each dimension into 100 equal segments, ranging from -10 to 10. As for the action space, we divided the values of each dimension into 50 equal segments, ranging from -1 to 1. In the case of Hopper, with its 3D action dimension, visualizing it similarly to the 1D action dimension in the Inverted Double Pendulum posed a challenge. Consequently, in the 3D plots, we assigned each axis to one of the action dimensions and utilized color to indicate the $Q$-value, as shown in Fig. 5. Additionally, in NTK analysis, representing the relationships between the reference action and all quantized actions within a single graph is challenging. We marked high-NTK regions in gray, where the normalized NTK values are greater than 0.9.

## E.2   $Q$-Network Weights Visualization

Figure 9, illustrating the neural network's learned weights for actions in two distinct datasets, provides a compelling visual representation of the over-generalization results presented in Figure 5. Specifically, the figure displays the first layer's weight matrix $W_1$ from a two-layer MLP $Q$-network trained on the Inverted Double Pendulum using Implicit $Q$-Learning (IQL). This network is defined as $Q(s, a) = W_2 \text{ReLU}(W_1(s, a) + b_1) + b_2$, where $W_1$ is a key focus due to its direct interaction with the concatenated state and action inputs. The dimensions of the weight matrix $W_1$ are $32 \times (\dim(\mathcal{S}) + \dim(\mathcal{A}))$, where $\dim(\mathcal{S}) = 11$ and $\dim(\mathcal{A}) = 1$ represent the dimensions of the state and action spaces, respectively. The figure contrasts the learned weights in datasets with different action spreads and the diversity of the $Q$-values: a narrow spread (expert dataset) and a wider spread (medium dataset).

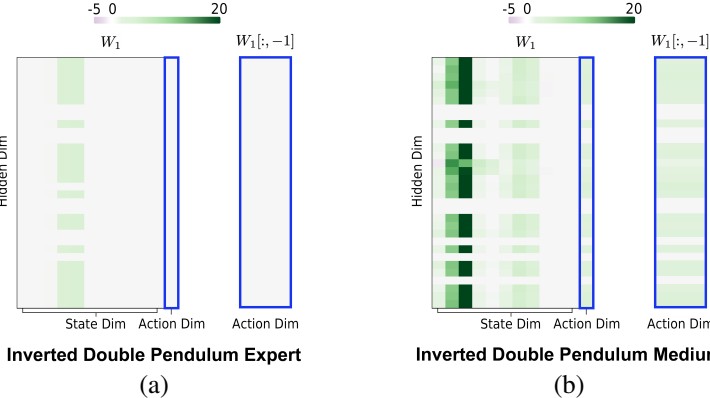

**Inverted Double Pendulum Expert**

(a)

**Inverted Double Pendulum Medium**

(b)

Figure 9: The first layer's weight matrix $W_1$ of a two-layer MLP $Q$-network trained on the Inverted Double Pendulum using IQL. The matrix dimensions are $32 \times (\dim(\mathcal{S}) + \dim(\mathcal{A}))$. For the expert dataset, the action-related elements of $W_1$ are learned as zero, indicating the network's training to not differentiate actions across all states.

For the expert dataset (Fig. 9(a)), the action-related elements of $W_1$ (the last column elements) are learned as zero. This intriguing result indicates that the network, during its training, learns not to differentiate between actions across all states, leading to uniformly flat $Q$-values for all actions. Such behavior is characteristic of datasets with a narrower action spread, where the actions are more clustered and coherent. The network's tendency to not distinguish between different actions in such a dataset is a direct consequence of the limited diversity, requiring less differentiation in the action representation.

In contrast, for the medium dataset (Fig. 9(b)), which represents a wider action spread, the action-related elements of $W_1$ show variation. This variation signifies that the network has learned to differentiate between actions to a greater extent, a necessity in a dataset where actions are more diverse and dispersed. The network's capacity to distinguish between various actions and assign different levels of importance to each reflects the need for a more nuanced understanding of the action space in datasets with a wider spread.

This visual evidence from the learned weights substantiates our understanding of how neural networks adapt their learning based on the diversity in the action space of the dataset. In datasets with narrower action spreads with similar $Q$-values, the network learns a more uniform approach towards different actions, while in those with wider spreads with diverse $Q$-values, it adopts a more differentiated and discerning strategy. This adaptive learning aligns with the principles of regression demonstrating the network's response to the diversity and distribution of actions in the training data.

### E.3  Extended NTK Visualization

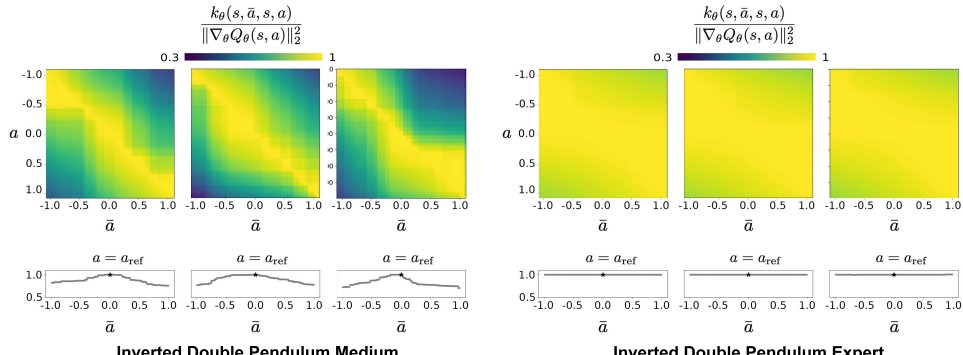

**Inverted Double Pendulum Medium**          **Inverted Double Pendulum Expert**

Figure 10: Normalized NTKs $k_\theta(s, \bar{a}, s, a)/\|\nabla_\theta Q_\theta(s, a)\|_2^2$ for three different fixed states from each dataset and for all reference action $a \in \mathcal{A}$ and contrastive action $\bar{a} \in \mathcal{A}$. The figures below illustrate the cross sections of figures above at $a = a_{\mathrm{ref}} = 0.0$.

In Fig. 5 (a) and (b) in Section 3.2, we visualize the NTK for a fixed state and a reference action $a_{\text{ref}}$ at zero in the Inverted Double Pendulum environment. Here, we present the extended results with three different fixed states and a varying reference action across the action space $a_{\text{ref}} \in [-1.0, 1.0]$. In Fig. 10, the diagonal symmetry of the normalized NTK as a function of action distance is observed. Generally, in the medium dataset, the NTK is high between two actions that are close to each other and low between actions that are far apart. However, in the expert dataset, this distinction becomes blurred, regardless of the proximity of the actions.

## E.4  Action Distributions

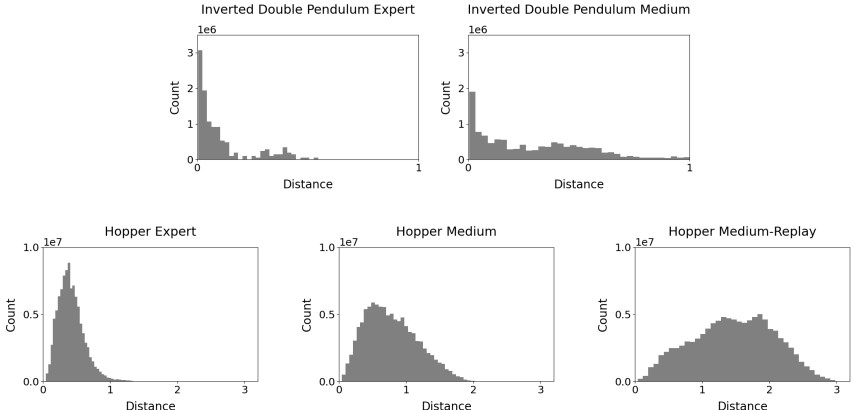

Figure 11: The average L2 distance between different actions within each quantized state in the Inverted Double Pendulum and MuJoCo Hopper environments. All histograms are plotted with 50 bins.

In this subsection, we examine the action distributions of datasets within the Inverted Double Pendulum and MuJoCo Hopper environments. The continuous nature and multi-dimensionality of these action spaces pose significant challenges for directly visualizing the exact action distributions. To address this, we define the average action distribution spread of a dataset $\mathcal{D}$, quantified as the expectation over the states of the dataset as:

$$H(\mathcal{D}) := \mathbb{E}_{s \in \mathcal{D}} \left[ \mathbb{E}_{a, \bar{a} \in \mathcal{D}(s)} \left[ \|a - \bar{a}\|_2 \right] \right]. \tag{8}$$

We then visualize the distribution of the L2 distance among all actions within each quantized state across the dataset using Eq. (8). This approach is based on the characteristics of the action dimensions in both the Inverted Double Pendulum and the Hopper, which are bounded between -1 and 1. For action distribution visualization, we maintain the same state and action quantization as outlined in Appendix E.1.

Fig. 11 presents the results. As depicted in this figure, in both the Inverted Double Pendulum and Hopper environments, the expert datasets exhibit a small average L2 distance between actions coexisting in the same quantized state. This indicates a denser clustering of actions within these datasets, which is linked to high-return datasets typically exhibiting a more concentrated action distribution on average since they primarily perform exploitation actions instead of exploration actions.

## F  Calculating $R^*$

In Section 4.1, we define the QCS weight $w(R(\tau))$ as $\lambda \cdot (R^* - R(\tau))$, where $R^*$ is the optimal return for the task. For calculating $R^*$, we consider the two methods described below.

(i) **Set $R^*$ with the optimal return for the environment.** In our experiments, we set $R^*$ for the environments with optimal returns as follows: Hopper ($R^* = 3500$), Walker2d ($R^* = 5000$), Halfcheetah ($R^* = 11000$), and AntMaze ($R^* = 1$). Note that prior RCSL algorithms such as Decision Transformer [10] and RvS [12] used predefined $R^*$ for target RTG conditioning

during inference. Therefore, using $R^*$ based on the optimal return introduces no additional assumptions compared to previous RCSL methods. As noted in Appendix J.2, QCS does not use $R^*$ for target RTG conditioning but instead relies on the maximum trajectory return, requiring only one $R^*$ per algorithm.

(ii) **Set $R^*$ to the maximum trajectory return within the dataset.** An alternative approach for setting $R^*$ is to use the maximum trajectory return from the dataset. When obtaining the true optimal return from the environment is challenging, the maximum trajectory return can serve as an approximation. Table 6 presents additional results using this method for $R^*$.

Table 6: QCS performance with $R^*$ as the optimal environment and maximum dataset returns.

| Dataset | QCS (optimal env return) | QCS (max dataset return) |
|---|---|---|
| halfcheetah-medium | 59.0 ± 0.4 | 55.2 ± 0.5 |
| hopper-medium | 96.4 ± 3.7 | 97.1 ± 3.0 |
| walker2d-medium | 88.2 ± 1.1 | 87.4 ± 2.1 |
| halfcheetah-medium-replay | 54.1 ± 0.8 | 52.1 ± 0.7 |
| hopper-medium-replay | 100.4 ± 1.1 | 99.8 ± 1.2 |
| walker2d-medium-replay | 94.1 ± 2.0 | 90.6 ± 3.2 |

As shown in Table 6, setting $R^*$ with the optimal environment return is slightly better than setting it with the maximum dataset return, but setting it with the maximum dataset return still outperforms the baselines. Therefore, we propose using the optimal environment return for $R^*$; however, when it is hard to determine, using the maximum dataset return can be a good alternative.

## G  More Experiments Results

### G.1  Additional Performance Comparison

In addition to the performance comparison in the MuJoCo and AntMaze domains, as discussed in Section 6.2, we also compare the performance of QCS in the Adroit domain using extensive baselines, similar to those mentioned in Section 6.2. Since there are no reported results for TD3+BC [14], SQL [52], RvS [12], QDT [53], EDT [49], CGDT [43], ACT [16], POR [50] in the Adroit domain, we only compare with the value-based baselines (IQL [23], CQL [25]) and RCSL baselines (DT [10], DC [20], RvS [12]). For DT and DC, we evaluate the score using their official codebase.

Table 7 displays the performance of QCS alongside the baseline performances. As indicated by the results, QCS-R outperforms other baselines in the Adroit Pen task. This outcome reiterates that QCS is a robust framework, excelling in a range of tasks with varying features. It also underscores findings from our earlier experiments, which demonstrate that strategically blending RCSL with $Q$-function can significantly enhance performance.

Table 7: The performance of QCS and baselines in the Adroit domain. The boldface numbers denote the maximum score.

|  | Value-Based Method | | RCSL | | Ours |
|---|---|---|---|---|---|
| Dataset | IQL | CQL | DT | DC | QCS-R |
| pen-human | 71.5 | 37.5 | 62.9 | 74.2 | **83.9** ± 10.2 |
| pen-cloned | 37.3 | 39.2 | 28.7 | 50.0 | **66.5** ± 9.5 |
| average | 54.4 | 38.4 | 45.8 | 62.1 | **75.2** |

### G.2  Comparison with FamO2O

FamO2O [42] is an offline-to-online RL method that facilitates state-adaptive balancing between policy improvement and constraints. During offline pre-training, it develops a set of policies with various balance coefficients. In the subsequent online fine-tuning phase, FamO2O determines the most suitable policy for each state by selecting the corresponding balance coefficient from this set. The major difference between QCS and FamO2O is that FamO2O additionally uses $10^6$ online samples to find a suitable balance coefficient, while QCS only utilizes the offline dataset and adjusts the balance coefficient (QCS weight) based on the trajectory return. Moreover, unlike FamO2O,

which utilizes a state-adaptive balance coefficient, QCS is based on historical architecture and uses a sub-trajectory-adaptive balance coefficient. Although it is not a fair comparison between FamO2O, an offline-to-online algorithm, and QCS, a purely offline algorithm, we present the performance comparison to demonstrate the effectiveness of QCS even when compared with an offline-to-online algorithm.

Table 8: Performance Comparison between FamO2O and QCS in the MuJoCo domain.

| Dataset | FamO2O (offline-to-online) | QCS (offline) |
|---|---|---|
| halfcheetah-m | **59.2** | **59.0** ± 0.4 |
| hopper-m | 90.7 | **96.4** ± 3.7 |
| walker2d-m | 85.5 | **88.2** ± 1.1 |
| halfcheetah-m-r | 53.1 | **54.1** ± 0.8 |
| hopper-m-r | 97.6 | **100.4** ± 1.1 |
| walker2d-m-r | 92.9 | **94.1** ± 2. |
| halfcheetah-m-e | **93.1** | **93.3** ± 1.8 |
| hopper-m-e | 87.3 | **110.2** ± 2.4 |
| walker2d-m-e | 112.7 | **116.6** ± 2.4 |
| average | 85.8 | **90.3** |

Table 9: Performance Comparison between FamO2O and QCS in the AntMaze domain.

| Dataset | FamO2O (offline-to-online) | QCS (offline) |
|---|---|---|
| antmaze-u | **96.7** | 92.5 ± 4.6 |
| antmaze-u-d | 70.8 | **82.5** ± 8.2 |
| antmaze-m-p | **93.0** | 84.8 ± 11.5 |
| antmaze-m-d | **93.0** | 75.2 ± 11.9 |
| antmaze-l-p | 60.7 | **70.0** ± 9.6 |
| antmaze-l-d | 64.2 | **77.3** ± 11.2 |
| average | 79.7 | **80.4** |

# H  More Ablation Studies

## H.1  Comparing Assistance from Actor-Critic Learned $Q$-Values

To compare the performance of QCS using $Q$-function learned through actor-critic algorithms, we use representative actor-critic algorithms such as CQL [25] for benchmarking. As shown in Table 10, the performance of CQL-aided QCS generally improved compared to the original CQL, but it does not match the performance of IQL-aided QCS for the MuJoCo domain. The reason can be attributed to two factors: (1) the $Q$-function may be bounded by the actor's representation ability, and (2) CQL might impose excessive conservatism on the $Q$-function. Moreover, in the case of `antmaze-umaze-diverse`, IQL-aided QCS underperforms CQL, but CQL-aided QCS outperforms CQL. Since QCS is a general framework that proposes a new combination of RCSL and $Q$-function on trajectory return, there is a wide range of potential integrations of RCSL and offline $Q$-learning methods. The most impactful aspect will differ depending on the characteristics of each RCSL and $Q$-learning method when combined. Exploring this would be an interesting research area, which we leave as future work.

Table 10: The performance of CQL, CQL-aided QCS, and IQL-aided QCS. The dataset names are abbreviated as follows: `medium` as 'm', `medium-replay` as 'm-r', and `umaze-diverse` as 'u-d'.

| Dataset | CQL | CQL-aided QCS | IQL-aided QCS |
|---|---|---|---|
| mujoco-medium | 58.3 ± 1.2 | 68.1 ± 1.5 | **81.2** ± 1.8 |
| mujoco-medium-replay | 72.6 ± 4.1 | 75.7 ± 5.8 | **82.9** ± 1.3 |
| antmaze-umaze-diverse | 84.0 | **85.2** | 82.5 |

## H.2  Impact of Base Architecture and Conditioning

In Section 4.3, we discussed QCS variants with different base architectures and conditioning. To assess the impact of these on performance, we conducted additional comparisons between QCS implementations with and without conditioning across three base architectures: DT, DC, and MLP. Table 11 reveals that the choice of base architecture does not significantly impact performance, except for the Adroit Pen. However, conditioning proves to be particularly beneficial for complex tasks and datasets with diverse trajectory optimality. Generally, we found that the DC base architecture is advantageous.

Table 11: Comparison of the base architecture of QCS and the ablations on conditioning. For the MuJoCo and Adroit domains, we utilize QCS-R, and for the AntMaze domain, we utilize QCS-G for evaluation. The dataset names are abbreviated as follows: `medium` to 'm', `medium-replay` to 'm-r', `medium-expert` to 'm-e', `umaze` to 'u', `umaze-diverse` to 'u-d', `medium-play` to 'm-p', `medium-diverse` to 'm-d', `large-play` to 'l-p', and `large-diverse` to 'l-d'. The boldface number represents the higher value when comparing the base architecture.

| Dataset | DT-based | DC-based | MLP-based |
|---|---|---|---|
| halfcheetah-m | 58.7 | **59.0** | 57.2 |
| hopper-m | 91.2 | **96.4** | 92.4 |
| walker2d-m | 85.4 | **88.2** | **88.6** |
| halfcheetah-m-r | 53.7 | **54.1** | 53.2 |
| hopper-m-r | 99.1 | 100.4 | **102.4** |
| walker2d-m-r | 90.9 | **94.1** | 93.3 |
| halfcheetah-m-e | **94.4** | 93.3 | 84.0 |
| hopper-m-e | **110.2** | **110.2** | **110.4** |
| walker2d-m-e | 115.4 | **116.6** | 115.4 |
| antmaze-u | 89.6 | 92.5 | **94.2** |
| antmaze-u-d | 72.3 | **82.5** | 78.7 |
| antmaze-m-p | 75.2 | **84.8** | 82.1 |
| antmaze-m-d | 72.1 | 75.2 | **80.1** |
| antmaze-l-p | 66.2 | **70.0** | 67.7 |
| antmaze-l-d | 75.3 | **77.3** | 68.9 |
| pen-human | 76.8 | **83.9** | 59.4 |
| pen-cloned | 40.2 | **66.5** | 44.0 |

| Dataset | DC-based Condition X | DC-based Condition O |
|---|---|---|
| halfcheetah-m | **58.7** | 59.0 |
| hopper-m | 83.3 | **96.4** |
| walker2d-m | 83.9 | **88.2** |
| halfcheetah-m-r | 53.6 | **54.1** |
| hopper-m-r | 76.6 | **100.4** |
| walker2d-m-r | 90.8 | **94.1** |
| antmaze-m-p | 80.3 | **84.8** |
| antmaze-m-d | 71.2 | **75.2** |
| antmaze-l-p | 41.2 | **70.0** |
| antmaze-l-d | 33.0 | **77.3** |
| pen-human | 60.7 | **83.9** |
| pen-cloned | 36.0 | **66.5** |

# I Training Curves

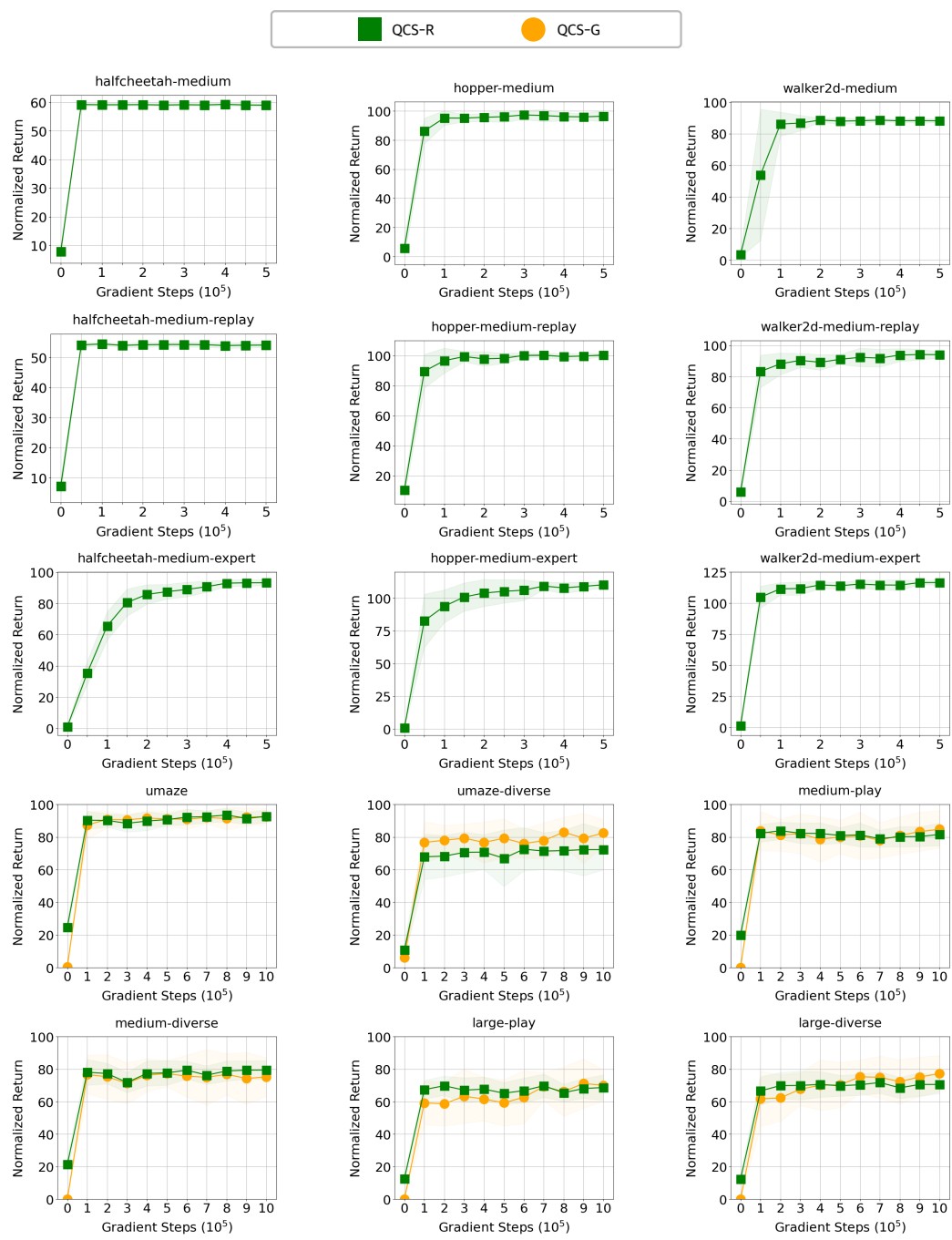

Figure 12: Training curves of QCS-R and QCS-G in the MuJoCo and AntMaze domains.

## J    Implementation and Hyperparameters

### J.1    Training the $Q$-Function

QCS utilizes $Q$-aid, where the $Q$-function is learned through IQL [23], using the open-source implementation of IQL (`https://github.com/Manchery/iql-pytorch`) and following the common hyperparameters recommended by the authors, as listed in Table 12. For AntMaze, we set the expectile to 0.8, whereas for other domains, we set it to 0.7. Moreover, inspired by RLPD [5] and SPOT [46], we employed Layer Normalization [4] and a larger discount 0.995 for the $Q$ and $V$ networks in AntMaze. For a fair comparison, we retrained IQL using these modified hyperparameters, and the results are shown in Table 13. Since this modified setting had a negative effect on IQL, we used the IQL performances from the IQL paper [23] for Table 3.

Table 12: Common hyperparameters for QCS $Q$-function training.

| Hyperparameter | Value |
|---|---|
| Optimizer | Adam [21] |
| Learning rate | 3e-4 |
| Batch size | 256 |
| Target update rate | 5e-3 |
| Hidden dim | 256 |
| Nonlinearity function | ReLU [2] |

Table 13: Comparison of the IQL performances reported in the IQL paper [23] with our results using modified hyperparameters.

| Dataset | IQL (reported in [23]) | IQL (modified hyper-params) |
|---|---|---|
| antmaze-u | 87.5 | 87.9 |
| antmaze-u-d | 67.2 | 38.7 |
| antmaze-m-p | 71.2 | 50.7 |
| antmaze-m-d | 70.0 | 45.3 |
| antmaze-l-p | 39.6 | 16.3 |
| antmaze-l-d | 47.5 | 9.3 |

### J.2    Policy Training

**Detailed description of the loss function.**

$$\mathcal{L}_\pi^{\text{QCS}}(\phi) = \mathbb{E}_{\mathcal{B}\sim\mathcal{D}}\left[\frac{1}{B}\sum_{i=1}^{B}\frac{1}{K}\sum_{j=0}^{K-1}\underbrace{\left\|a_{t_i+j}^{(i)} - \pi_\phi\left(\tau_{t_i:t_i+j}^{(i)}\right)\right\|_2^2}_{\text{RCSL}} - \frac{\lambda\cdot\left(R^* - R(\tau^{(i)})\right)}{\bar{Q}_\theta^{\text{IQL}}}\underbrace{Q_\theta^{\text{IQL}}\left(s_{t_i+j}^{(i)}, \pi_\phi\left(\tau_{t_i:t_i+j}^{(i)}\right)\right)}_{Q\text{ Aid}}\right],$$

$$(9)$$

where each component of the loss function is as follows:

- The batch sampled over the entire dataset $\mathcal{D}$ (e.g., `hopper-medium`):

$$\mathcal{B} = \left\{\tau_{t_1:t_1+K-1}^{(1)}, \dots, \tau_{t_B:t_B+K-1}^{(B)}\right\}, B = |\mathcal{B}|.$$

- $i$-th sub-trajectory in the batch for $i = 1, \dots, B$:

$$\tau_{t_i:t_i+K-1}^{(i)} = \left(\hat{R}_{t_i}^{(i)}, s_{t_i}^{(i)}, a_{t_i}^{(i)}, \dots, \hat{R}_{t_i+K-1}^{(i)}, s_{t_i+K-1}^{(i)}\right).$$

- Dataset-level $Q$-normalizer:

$$\bar{Q}_\theta^{\text{IQL}} = \frac{1}{|\mathcal{D}|}\sum_{(s,a)\in\mathcal{D}}Q_\theta^{\text{IQL}}(s,a),$$

i.e., the dataset-level $Q$-normalizer $\bar{Q}_{\text{IQL}}$ is the mean of the $Q$-value for all samples in the dataset.

**Implementations.** After training the $Q$-function, we train our policy with three different base architectures: DT [10], DC [20], and MLP. For DT-based QCS, we utilize the official DT codebase (`https://github.com/kzl/decision-transformer`) for our implementation. Similarly, for DC-based QCS, we use the official DC codebase (`https://github.com/beanie00/Decision-ConvFormer`) for our implementation.

**Hyperparameters.** For the AntMaze domain, we used $10^6$ gradient steps, and for the other domains, we used $5 \times 10^5$ gradient steps for training the policy. For all domains and base architectures, QCS uses a dropout rate of 0.1, ReLU as the nonlinearity function, a weight decay of 1e-4, and a LambdaLR scheduler [33] with a linear warmup of $10^4$ gradient steps. In addition, we use a context length $K$ of 20 for DT-based QCS, 8 for DC-based QCS, and 1 for MLP-based QCS. We found that the impact of action information and positional embedding on performance was negligible, so we excluded them from training. In QCS-R, we set our target RTG to the highest trajectory return in the dataset. For the MuJoCo and Adroit domains, we evaluate the target RTG at double this value. In the AntMaze domain, we test it at 100 times the value. This method aims to leverage the RTG generalization effect observed by Kim et al. [20]. We then report the best score achieved across the two target RTGs. From Table 14 to 15, we provide detailed hyperparameter settings for actor training.

Table 14: Per-domain hyperparameters of DT-based QCS and DC-based QCS.

| Hyperparameter | MuJoCo | AntMaze | Adroit |
| --- | --- | --- | --- |
| Hiddem dim | 256 | 512 | 128 |
| # layers | 4 | 3 | 3 |
| Batch size | 64 | 256 | 64 |
| Learning rate | 1e-4 | 3e-4 | 3e-4 |

Table 15: Per-domain hyperparameters of MLP-based QCS.

| Hyperparameter | MuJoCo | AntMaze | Adroit |
| --- | --- | --- | --- |
| Hiddem dim | 1024 | 1024 | 256 |
| # layers | 3 | 4 | 3 |
| Batch size | 64 | 256 | 64 |
| Learning rate | 1e-4 | 3e-4 | 3e-4 |

**QCS Weight Relative to Trajectory Return.** Our analysis suggests setting the QCS weight $w(R(\tau))$ as a continuous, monotone-decreasing function of the trajectory return $R(\tau)$. We explored various functional forms, including linear, quadratic, and exponential decay, but found that a simple linear decay $w(R(\tau)) = \lambda (R^* - R(\tau))$ suffices. In addition, we found that for some datasets, clipping $w(R(\tau))$ to a minimum of 10 is beneficial, particularly for `walker2d-medium-expert` and QCS-R AntMaze, except `umaze-diverse`. The choice of $\lambda$ for each dataset is presented in Table 16 to 18.

Table 16: $\lambda$ on MuJoCo.

| Dataset | $\lambda$ |
| --- | --- |
| halfcheetah-medium | 1 |
| halfcheetah-medium-replay | 1 |
| halfcheetah-medium-expert | 0.5 |
| hopper-medium | 0.5 |
| hopper-medium-replay | 0.5 |
| hopper-medium-expert | 0.5 |
| walker2d-medium | 0.5 |
| walker2d-medium-replay | 1 |
| walker2d-medium-expert | 1 |

Table 17: $\lambda$ on AntMaze.

| Dataset | $\lambda$ |
| --- | --- |
| antmaze-umaze | 0.2 |
| antmaze-umaze-diverse | 0.05 |
| antmaze-medium-play | 0.2 |
| antmaze-medium-diverse | 0.2 |
| antmaze-large-play | 0.2 |
| antmaze-large-diverse | 0.2 |

Table 18: $\lambda$ on Adroit.

| Dataset | $\lambda$ |
| --- | --- |
| pen-human | 0.01 |
| pen-cloned | 0.01 |

# K    Training Time

We compare QCS training time complexity with IQL [23], CQL [25], and QDT [53]. QCS requires a pre-trained Q learned using the IQL method, while QDT requires a pre-trained Q learned using the CQL method. Therefore, the time was measured by incorporating the Q pretraining time for both algorithms.

The training times for IQL, CQL, QDT, and QCS are as follows: IQL - 80 min, CQL - 220 min, QDT - 400 min, and QCS - 215 min.

The results show that QCS takes longer than IQL but has a total time similar to CQL. Notably, compared to QDT, which requires CQL pretraining, QCS can be trained in nearly half the time but demonstrates superior performance to QDT as shown in our main results in Table 2.

## L   Limitations

In this paper, we leveraged the complementary relationship between RCSL and $Q$-function over-generalization to determine the QCS weight as a linear function of the trajectory return, which is readily obtainable from the dataset. This approach was tested on MuJoCo, AntMaze, and Adroit, where it showed promising results. However, depending on the task, a more advanced method that can efficiently evaluate $Q$-functions's over-generalization and provide appropriate $Q$-aid might be necessary. Additionally, this method entails the extra burden of pre-training the $Q$-function.

## M   Borader Impacts

This research is centered on enhancing the strengths of two promising approaches in the field of offline reinforcement learning: RCSL and value-based methods. By overcoming each of their limitations and creating better trajectories than the maximum quality trajectories of existing datasets, it contributes to the advancement of offline reinforcement learning. As foundational research in machine learning, this study does not lead to negative societal outcomes.

