# OpenReview forum: "Adaptive $Q$-Aid for Conditional Supervised Learning in Offline Reinforcement Learning"
_NeurIPS.cc/2024/Conference — NeurIPS 2024 poster_

### Official Review · Reviewer_Ndxq · 2024-06-25

**Soundness:** 3
**Presentation:** 4
**Contribution:** 3
**Rating:** 7
**Confidence:** 4

**Summary:**

This paper proposes an offline reinforcement learning framework by combining return-conditioned supervised learning with in-sample Q-learning. By demonstrating their relative merits on offline datasets with different behavior policies, an overall loss function is designed to integrate Q assistance into RCSL. Extensive experiments empirically validate its superior performance with compared baselines on several domains and datasets.

**Strengths:**

1.	The studied problem is important. Empowering RCSL with stitching ability is crucial to facilitate its capability for pursuing optimal policies.
2.	The analysis and experiments of RCSL and Q-learning on datasets with optimal and suboptimal policies provide valuable insights. Based on that, the proposed QCS method can be naturally optimized on the proposed loss function by considering conditional supervising learning and maximizing Q-function simultaneously.
3.	Empirical experimental results are promising with a bunch of baselines on several datasets and domains.

**Weaknesses:**

1.	The definition of degree of Q-aid as w(R($\tau$)) is not completely reasonable. As described, it should be defined by the degree of optimality of the behavior policy used to generate this sequence. However, a sequence with a higher return does not necessarily indicate it was generated by a superior policy, even from a statistical perspective. This definition only makes sense if the initial states of the compared trajectories are the same.
2.	The analysis of the Q-Greedy Policy appears somewhat redundant. The focus of this paper is on integrating Q-learning into GCSL, rather than addressing challenges in Q-learning.
3.	Conversely, the experimental section seems overly brief. I suggest moving more experimental details from the appendix to the main paper.

**Questions:**

All my questions are listed in the weakness part.

**Limitations:**

No limitation issues.

---

> ### Author Rebuttal · Authors · 2024-08-06
>
> Thank you for your positive comments on the strengths of our work. Your valuable suggestions will help us further improve and clarify our work.
>
> &nbsp;
>
> ### **W1. The definition of degree of Q-aid.**
>
> Thank you for pointing out a good point. We have dealt with environments where the deviations in the initial state are marginal to affect the scale of the returns significantly, but in practical scenarios, differences may arise. In such cases, a higher trajectory return may not always indicate optimality. Considering how to set the weights in a more practical setting with significant differences in the initial state could be a good follow-up work.
>
> &nbsp;
>
> ### **W2. Regarding writing.**
>
> As the reviewer mentioned, the focus of our work is combining Q-learning with RCSL. To develop an algorithm for this purpose, we first (1) experimentally demonstrated that the Q-greedy policy and RCSL can be complementary depending on the quality of the dataset, and (2) analyzed the reasons for this through an examination of Q generalization. Based on this analysis, we determined the QCS weight and developed the QCS algorithm. Therefore, we believe that discussing the Q-greedy policy is necessary to explain the need for and development process of our algorithm. However, we will improve the writing to make this point brief and clear. However, we will improve the writing to make this point brief and clear. Additionally, we will refine our manuscript to include more experimental details in the main section.

---

### Official Review · Reviewer_fTMz · 2024-07-09

**Soundness:** 3
**Presentation:** 3
**Contribution:** 3
**Rating:** 7
**Confidence:** 4

**Summary:**

Authors of this paper introduce a new algorithm called Q-aided Conditional Supervised Learning (QCS) for offline DRL scenario, which combines the stability of return-conditioned supervised learning (RCSL) and the stitching ability of Q-functions. The primary contributions are 1). Identify the strength and weakness of RCSL and Q-learning in different experimental settings and 2) propose a new approach to combine the them to achieve better performance.

**Strengths:**

1. The paper is well written. The paper has a logical structure on the problem definition. It first demonstrates the strength and weakness of RCSL and Q-learning in different offline settings through experimental results and then explain the reasons through simple toy examples to help understanding. Finally, it propose the novel approach QCS based on the reasoning above.

2. The paper has rigorous experimental set-up and SOTA performance. The proposed algorithm is tested on various offline D4RL benchmarking datasets and demonstrate SOTA performance across the datasets.

3. Very interesting study on the strength and weakness of RCSL and Q-learning in different experimental settings, especially the discovery of Q-learning over generalization problem.

**Weaknesses:**

1. If I understand correctly, QCS requires a pre-trained Q-function using IQL, while the experimental results on D4RL benchmarking seems very impressive, there appears to be a lack of detailed discussion concerning the computational workload.

2. The weight term (which is important and essential) in the algorithm is dynamic and depends on R*, the optimal return of the task. Calculating R* across environments and dataset settings can be a bit tricky and discussion on how calculation is performed is a also a bit limited and unclear.

3. Different benchmark experiments seem to require different hyper-parameter $\lambda$ values to achieve SOTA performance. Not sure how sensitive QCS is to $\lambda$.

**Questions:**

1. For calculating R*, for each update using equation (3), do you need to calculate R* for each trajectory? Could you provide more detailed information on how it is calculated?

2. I think the hyper-parameter K has very limited discussion on its impact on the performance. Do you perform any ablation studies on how K will affect the model performance?

3. How sensitive is the QCS algorithm to the scaling factor $\lambda$ ? I noticed that, for one environment with different offline settings, (for example, halfcheetah-medium-replay and halfcheetah-medium-expert), OCS uses different $\lambda$ values. Does one fixed $\lambda$ work well within the same environments? I think this is important for understanding the adaptability and robustness of the method across different settings.

**Limitations:**

The paper briefly mentions the potential limitation of using a simple linear function to combine RCSL and Q-function.

---

> ### Author Rebuttal · Authors · 2024-08-06
>
> Thank you for the detailed comments that help clarify our algorithm and for the positive feedback on our work. As suggested, we have included further explanations and experimental results to enhance the significance of our study. Due to the rebuttal word limit, **we have addressed the response regarding context length in the global response G5.**
>
> &nbsp;
>
> ### **W1. Regarding complexity.**
>
> We compare QCS training time complexity with IQL, CQL, and QDT [Ref.8]. QCS requires a pre-trained Q learned using the IQL method, while QDT requires a pre-trained Q learned using the CQL method. Therefore, the time was measured by incorporating the Q pretraining time for both algorithms.
>
> The training times for IQL, CQL, QDT, and QCS are as follows:
> IQL - 80 min, CQL - 220 min, QDT - 400 min, and QCS - 215 min.
>
> The results show that QCS takes longer than IQL but has a total time similar to CQL. Notably, compared to QDT, which requires CQL pretraining, QCS can be trained in nearly half the time but demonstrates superior performance to QDT as shown in our main results in Table 2.
>
> &nbsp;
>
> [Ref.8] Yamagata, Taku, et al. "Q-learning decision transformer: Leveraging dynamic programming for conditional sequence modeling in offline rl." ICML 2023.
>
> &nbsp;
>
> ### **W2/Q1. Regarding calculating R***
>
> The detailed explanation of how to obtain R$^*$ is as follows. We propose two ways of calculating R$^*$, and we will add this explanation to our manuscript.
>
> * (1) Set R$^*$ with the optimal return for the environment.
>
> For our main experiments, we set R$^*$ for the environments with the optimal return for the environment as follows: Hopper - 3500, Walker2d - 5000, Halfcheetah - 11000, and Antmaze - 1. Note that prior RCSL algorithms, such as Decision Transformer [Ref.8] or RvS [Ref.7], used predefined R$^*$ for target RTG conditioning at the inference stage. Therefore, setting R$^*$ with optimal return requires no further assumptions compared to prior RCSL works. As mentioned in Appendix J.2 of our manuscript, for target RTG conditioning, QCS does not use R$^*$ but instead uses the maximum trajectory return, only requiring one R$^*$ per algorithm.
>
> * Set R$^*$ to the maximum trajectory return within the dataset.
>
> Another way of setting R$^*$ is to use the maximum trajectory return within the dataset. In situations where obtaining the optimal environment return is difficult, we can infer the optimal return using the maximum trajectory return. Therefore, in Table R4 we provide additional results using the maximum trajectory return as R$^*$.
>
> &nbsp;
>
> **[Table R4. QCS performance when R\* is the maximum dataset return and the optimal environment return.]**
>
> | Dataset                   | QCS (optimal env return) | QCS (max dataset return) |
> |---------------------------|--------------------------|--------------------------|
> | halfcheetah-medium        | 59.0±0.4                 | 55.2±0.5                 |
> | hopper-medium             | 96.4±3.7                 | 97.1±3.0                 |
> | walker2d-medium           | 88.2±1.1                 | 87.4±2.1                 |
> | halfcheetah-medium-replay | 54.1±0.8                 | 52.1±0.7                 |
> | hopper-medium-replay      | 100.4±1.1                | 99.8±1.2                 |
> | walker2d-medium-replay    | 94.1±2.0                 | 90.6±3.2                 |
>
>
> As shown in Table R4, setting R$^*$ with the optimal environment return is slightly better than setting it with the maximum dataset return, but setting it with the maximum dataset return still outperforms the baselines. Therefore, we propose using the optimal environment return for R$^*$; however, when it is hard to determine, using the maximum dataset return can be a good alternative.
>
> &nbsp;
>
> [Ref.7] Emmons, Scott, et al. "RvS: What is Essential for Offline RL via Supervised Learning?." ICLR 2022.
>
> [Ref.8] Chen, Lili, et al. "Decision transformer: Reinforcement learning via sequence modeling." NeurIPS 2022.
>
> &nbsp;
>
> ### **W3/Q3. Impact of the QCS weight $\lambda$.**
>
> We wish to direct the reviewer's attention to Table 9 in Appendix H.2 of our original manuscript, which investigates the effect of $\lambda$ by varying it from 0.2 to 1.5. We additionally test the QCS with $\lambda$ 0.5 and 1 with MuJoCo medium-expert datasets and the result can be seen in Table R5.
>
> As shown in Table 9 and R5, except for the walker2d-medium and halfcheetah-medium-expert, we found that even the smallest values achieved with changing $\lambda$ either matched or surpassed the performance of existing value-based methods and RCSL's representative methods, including IQL, CQL, DT, DC, and RvS. This demonstrates QCS's relative robustness regarding hyperparameters. For walker2d-medium and halfcheetah-medium-expert, we found that when $\lambda$ exceeds the initial setting of 0.5, performance begins to decrease. We anticipate that further research will address these issues.
>
> Overall, setting $\lambda$ to 0.5 consistently produces good performance, even though it results in performance degradation of around 2 points in some environments. **Compared to many offline RL algorithms that require tuning more than 10 sets of hyperparameters depending on the environment, achieving stable performance with only one or two hyperparameter adjustments is a major strength of QCS.**
>
> &nbsp;
>
> **[Table R5. Impact of the QCS weight $\lambda$ in MuJoCo medium-expert datasets.]**
>
> | Dataset                   | $\lambda$=0.5           | $\lambda$=1             |
> |---------------------------|-------------------------|-------------------------|
> | halfcheetah-medium-expert | 93.3±1.8                | 84.6±4.2                |
> | hopper-medium-expert      | 110.2±2.4               | 110.8±2.8               |
> | walker2d-medium-expert    | 117.4±2.0               | 116.6±2.4               |

---

### Official Review · Reviewer_K7ih · 2024-07-13

**Soundness:** 2
**Presentation:** 2
**Contribution:** 2
**Rating:** 6
**Confidence:** 2

**Summary:**

Offline reinforcement learning (RL) has advanced with return-conditioned supervised learning (RCSL) but still lacks stitching ability. Q-Aided Conditional Supervised Learning (QCS) combines RCSL's stability with the stitching capability of Q-functions, addressing Q-function over-generalization. QCS adapts Q-aid integration into RCSL's loss function based on trajectory returns, significantly outperforming RCSL and value-based methods in offline RL benchmarks. This innovation pushes the limits of offline RL and fosters further advancements.

**Strengths:**

- it is well-written and fluent.

- They brought clear definitions which make it easy to follow the context.

- They compared with different baselines.

**Weaknesses:**

The innovation of the works is limited.

**Questions:**

- how the subgoals are chosen? Is not it complicated to choose them in a more complex environment?

**Limitations:**

The conditioning part seems to be controled by the algorithm which makes it hard to use in any environment.

---

> ### Author Rebuttal · Authors · 2024-08-06
>
> Thank you for providing comments and positive feedback on our work. We hope that our response below will clarify our algorithm and innovations.
>
> &nbsp;
>
> ### **W1. The innovation of QCS.**
>
> We summarize the innovations of QCS as follows:
> * We first analyze the strengths and weaknesses of RCSL and Q-learning in various dataset qualities and discover that RCSL and Q-learning can be complementary depending on the quality of the dataset.
> * Especially, we identify the Q-function over-generalization problem when performing Q-learning with an optimal quality dataset.
> * Based on the analysis of Q-function over-generalization, we propose a novel algorithm that combines RCSL and Q-learning, called QCS. QCS demonstrates superior performance compared to various recent and SOTA baselines.
>
> We want to emphasize that our analysis of the complementary conditions of RCSL and Q-learning, as well as the Q-function over-generalization problem, is novel and has not been addressed in previous works. The superiority of the QCS algorithm, which is based on these analyses, further confirms that our analyses are necessary and can lead to a deeper understanding within the offline RL community.
>
> &nbsp;
>
> ### **Q1. Regarding the subgoal selection.**
>
> For QCS-G, we follow the subgoal selection as proposed by RvS [Ref.7]. As per RvS, during the training phase, we randomly select a subgoal from among the states between the next state and the final state. For the evaluation phase, we prefix the subgoal as the goal state. We select subgoals based on the states present in the dataset, and there is no need for a special algorithm; we simply choose randomly among the next states, incurring no additional computational cost or difficulty.
>
> Moreover, as can be seen in Table 3 of our manuscript, the QCS-R score, which only utilizes reward information, shows performance comparable to QCS-G. This demonstrates that the QCS algorithm can achieve good performance even without subgoal selection.
>
> &nbsp;
>
> [Ref.7] Emmons, Scott, et al. "RvS: What is Essential for Offline RL via Supervised Learning?." ICLR 2022.

---

> > ### Comment · Reviewer_K7ih · 2024-08-09
> >
> > Thanks for the response.
> >
> > How randomly selected subgoals can always achieve to a good performance?

---

> ### Author Response · Authors · 2024-08-10
>
> Thank you for bringing up a good point. As we mentioned in our previous response, subgoal conditioning follows the prior work [Ref.7], and our explanation of the effect of the subgoal is summarized below.
>
> Let's first reconsider return conditioned supervised learning. In the case of return conditioning, the agent basically learns $a = f_\theta(s, \hat{R})$, where $\hat{R}$ is the return-to-go, i.e., sum of rewards received from the current timestep until the end of each episode. During training, the joint function $f_\theta$ of $(s, \hat{R})$ is learned with   various $\hat{R}$ values from good or bad trajectories in the dataset.  Basically we learn the  function $f_\theta$ on the $(s, \hat{R})$ plane hoping for generalization over a wide region of $(s, \hat{R})$ with various $(s,\hat{R})$ data points.  Later during the test phase, we set a high target return-to-go as $\hat{R}$ so that  the model will output an action
> $a$ for each $s$ with  high return-to-go. That is, we use $a=f_\theta(s, \hat{R}\_{target})$, a function  from $s$ to $a$, where $\hat{R}\_{target}$ is a fixed high target value.
>
> Similarly, subgoal conditioning case can be considered to learn a function $a = f_\psi(s, \hat{s}\_{goal})$, where the subgoal $ \hat{s}\_{goal}$ is a randomly-picked state  between the next step and  the episode end  in each episode.
> If $\hat{s}\_{goal}$ is from a bad episode, the produced action will be bad.  If $\hat{s}\_{goal}$ is from a good episode, the produced action will be good.
> But, again, our goal is to learn an entire function $a = f_\psi(s, \hat{s}\_{goal})$ generalized over a wide  joint region of $(s, \hat{s}\_{goal})$, where $\hat{s}\_{goal}$ ranges from good to bad (hoping for interpolation and extrapolation) so that in the later test phase, we use this function $a = f_\psi(s, \hat{s}\_{goal})$ generalized over a wide  joint region of $(s, \hat{s}\_{goal})$ by fixing $\hat{s}\_{goal}$ to be the ultimate goal of the task, which is known for each task.
>
> We hope our additional response has addressed all your concerns. If your concerns have been resolved or if you have any further questions, please feel free to let us know.
>
> &nbsp;
>
> [Ref.7] Emmons, Scott, et al. "RvS: What is Essential for Offline RL via Supervised Learning?." ICLR 2022.

---

> > ### Author Response · Authors · 2024-08-11
> > **Further clarification on previous response**
> >
> > Adding on to our previous response, random selection of subgoals occurs during training to learn a generalized function for various optimal and non-optimal subgoals.  However, during evaluation, instead of using random subgoals, the agent always uses the fixed ultimate goal of the task, which is known for each task. We hope this helps to further clarify your concerns regarding our subgoal conditioning method.

---

> > > ### Comment · Reviewer_K7ih · 2024-08-11
> > >
> > > Thanks for the response addressing my concern. I increase my score to 6.

---

> > > > ### Author Response · Authors · 2024-08-11
> > > >
> > > > We are pleased to hear that our response has resolved your concerns. Thank you once again for leaving a thoughtful comment and for dedicating your time and effort to reviewing our work.

---

### Official Review · Reviewer_WnjH · 2024-07-13

**Soundness:** 2
**Presentation:** 1
**Contribution:** 2
**Rating:** 3
**Confidence:** 3

**Summary:**

This submission proposes an algorithm that combines the stability of return-conditioned supervised learning (RCSL) with the stitching capability of Q-functions. The submission tests their algorithm in the MuJoCo domain with medium, medium-replay, and medium-expert datasets. The performance of the submission’s proposed method is higher on average in the MuJoCo domain.

**Strengths:**

The paper tests their proposal in the main benchmarks.

**Weaknesses:**

The HQIL algorithm [1] outperforms the proposed method of the submission. However, this is not mentioned, and further there is no comparison to HQIL.

Table 1 might need some standard deviations.

To be able to interpret Table 2 correctly, the table needs to have standard deviations of the previous methods as well.

POR method can also perform above 70 in the antmaze-large-diverse dataset.

Other studies also report results for antmaze-medium-replay dataset. Why did the authors omit this baseline when reporting results?

Table 8 again needs standard deviations.

The submission only compares their algorithm to old baselines. However, there are currently algorithms that outperforms the proposal of the submission.

Section 3.2. argues about something that is already expected/known to anyone that knows basic reinforcement learning, i.e. how the $Q$-learning update works. Should this section really occupy two pages in the main body of the paper?

*“QCS represents a breakthrough in offline RL,pushing the limits of what can be achieved and fostering further innovations.”*

This statement might be too strong to describe the submission's contributions

[1] HIQL: Offline Goal-Conditioned RL with Latent States as Actions, NeurIPS 2023.

**Questions:**

Why did the authors bold their algorithm while SQL is the highest performing algorithm for hopper-m-e in Table 2? (111.8>110.2)

**Limitations:**

Please see weaknesses.

---

> ### Author Rebuttal · Authors · 2024-08-06
>
> We appreciate the reviewer's efforts in providing constructive feedback to improve our work. **Our detailed response to the reviewer's comments is posted below and also in the global response G1-G4.**
>
> &nbsp;
>
> ### **W1. Comparison with HIQL.**
>
> Thank you for introducing the highly effective algorithm HIQL [Ref.2] for the goal conditioning task. RL tasks can be divided into two categories: return-maximizing tasks, which aim to earn the maximum return, like MuJoCo, and goal-reaching tasks, which aim to reach the goal with a higher success ratio, like Antmaze. HIQL specializes in goal-reaching tasks as it uses a hierarchical approach that generates high-valued subgoals with a higher-level policy and generates actions with a lower-level policy conditioned on those generated subgoals. HIQL achieves superior performance on Antmaze tasks, but due to the nature of the algorithm, which involves generating and conditioning on subgoals, it has limitations when applied to tasks like MuJoCo.
>
> In Table R2, we compare QCS-R and QCS-G with HIQL in the Antmaze medium and large datasets. QCS previously used a hidden dimension of 256 for training the Q-function, and we observed that HIQL uses a hidden dimension of 512. Therefore, we re-trained the Q-function with an increased hidden dimension of 512 and confirmed that this setting is beneficial for QCS. In Table R2, we mark the original QCS-R/QCS-G score as QCS-R(256)/QCS-G(256) and the new score as QCS-R(512)/QCS-G(512).
>
> As shown in Table R2, our new QCS score is comparable to HIQL, which is specialized for goal conditioning tasks, and particularly, QCS-R performs even better than HIQL. This new result reaffirms that our algorithm, despite being general, is effective even when compared to specialized algorithms.
>
> &nbsp;
>
> **[Table R2. Performance comparison between QCS and HIQL.**]
>
> | Dataset     | HIQL          | QCS-R(256)     | QCS-R(512)     | QCS-G(256)     | QCS-G(512)     |
> |-------------|---------------|----------------|----------------|----------------|----------------|
> | antmaze-m-p | 84.1±10.8     | 81.6±6.9       | **93.1±7.0**   | 84.8±11.5      | 88.5±6.3       |
> | antmaze-m-d | 86.8±4.6      | 79.5±5.8       | **88.7±5.4**   | 75.2±11.9      | 84.9±8.4       |
> | antmaze-l-p | 88.2±5.3      | 68.7±7.8       | **89.2±6.3**   | 70.0±9.6       | 84.2±10.5      |
> | antmaze-l-d | **86.1±7.5**  | 70.6±5.6       | 84.2±10.6      | 77.3±11.2      | 77.1±7.0       |
> | **average** | **86.3**      | 75.1           | **88.8**       | 76.8           | 83.7           |
>
> &nbsp;
>
> [Ref.2] Park, Seohong, et al. "Hiql: Offline goal-conditioned rl with latent states as actions." NeurIPS 2023.
>
> &nbsp;
>
> ### **W2. Recentness of Baselines.**
>
> We want to note that we compared baselines including very recent works such as ACT (AAAI 2024), CGDT (AAAI 2024), DC (ICLR 2024), FamO2O (NeurIPS 2023), EDT (NeurIPS 2023), QDT (ICML 2023), SQL (ICLR 2023). Especially since the NeurIPS 2024 submission deadline was May 22, considering this date, the works for AAAI 2024 and ICLR 2024 can be regarded as very recent, having been published less than three months ago.
>
> We continuously track the latest works, and after the NeurIPS submission deadline, we found a new work, Q-value Regularized Transformer(QT) [Ref.6] that proposes a new way of combining RCSL and Q-learning, published at ICML 2024. Note that according to NeurIPS regulations, QT is considered `Contemporaneous Work' because it appeared online after the submission. Since the work appeared after our submission, we are not obligated to compare it, but to help the reviewer better appreciate the quality of our work, we additionally compare QCS with QT in Table R3 in response W3 using the same evaluation metrics as QCS.
>
> [Ref.6] Hu, Shengchao, et al. "Q-value Regularized Transformer for Offline Reinforcement Learning." ICML 2024.
>
> &nbsp;
>
> ### **W3. Comparison with QT and POR with the same evaluation metric of QCS.**
> As mentioned in our manuscript Section 6.1, we report QCS as the **last running average score**. However, QT selects the best score throughout the whole training process (as confirmed through email with the QT author), which is not suitable for offline RL settings. Offline RL assumes a situation with limited online interaction, but finding the best score relies heavily on a significant amount of online interaction. Therefore, we re-ran the QT through the official QT code with author-recommended hyperparameters and verified the last running average score.
>
> Regarding POR, mentioned by the reviewer for the antmaze large score, we previously reported POR scores from the POR paper. We were unable to confirm the exact evaluation metric, but the authors shared the training curve of POR on the POR github. From this, we verified the last running average score of POR.
>
> To summarize, in Table R3 and Table R9 in the global response PDF, we compare QCS with QT and POR using the last running average score. As can be seen in Tables, QCS outperforms both POR and QT in MuJoCo and Antmaze domains, with POR's Antmaze large scores being below 70, demonstrating that QCS is a robust and high-performing algorithm.
>
> &nbsp;
>
> **[Table R3. Performance comparison between QCS (ours), POR, and QT. We evaluate each algorithm based on the last running average score.]**
>
> | Dataset        | POR (last)      | QT (last)      | QCS (ours, last)      |
> |----------------|-----------------|----------------|-----------------|
> | antmaze-u      | 87.9±4.0        | 51.3±15.7      | **92.5**±4.6        |
> | antmaze-u-d    | 65.5±6.0        | 57.9±9.6       | **82.5**±8.2        |
> | antmaze-m-p    | **84.6**±5.4        | 32.7±11.3      | **84.8**±11.5       |
> | antmaze-m-d    | **75.6**±4.8        | 22.7±23.6      | **75.2**±11.9       |
> | antmaze-l-p    | 56.5±5.2        | 0±0.0          | **70.0**±9.6        |
> | antmaze-l-d    | 63.0±4.0        | 0±0.0          | **77.3**±11.2       |
> | **average**    | 72.2        | 27.4       | **80.4**        |
>
> &nbsp;

---

> > ### Comment · Reviewer_WnjH · 2024-08-13
> >
> > I thank the authors for their response and providing the standard deviation results.
> >
> > The results reported in the original paper [1] for POR are different from what the submission reports. For instance, for the hopper-m dataset for POR the original paper reports 98.2 ±1.6, however the submission reports POR results as 78.6 ± 7.2 for the hopper-m dataset. Given that the performance of the submission’s proposed algorithm QCS-R is 96.4 ± 3.72, this means the submission does not outperform a prior algorithm POR. The way the results are reported in the submission is quite misleading and incorrect.
> >
> > [1] A Policy-Guided Imitation Approach for Offline Reinforcement Learning, NeurIPS 2022.
> >
> > Looking at the results reported in Table 9 in the attached pdf, in the hopper-m-e dataset the performance of QT is 108.2± 3.6, and the performance of QCS is 110.2± 2.41. For the halfcheetah-m-e dataset the performance of QT is 94.0±0.2 the performance of QCS 93.3±1.78. These results are within one standard deviation.
> >
> > *“When learning $Q_\theta$ from such limited information, where the values at the narrow action points are almost identical for each given state, it is observed that the learned $Q_\theta$ tends to be over-generalized to the OOD action region.”*
> >
> > Where was this observed? It would be good to have a reference here if this observation comes from prior work.
> >
> > How would the results look like for POR and QT if you were reporting results as the prior work did, i.e. selecting the best score throughout the whole training process as you confirmed with the QT author through email, instead of the last running average score? Before changing the metrics we report the results, perhaps we can provide results in both metrics to be able to provide robust comparisons with prior work.
> >
> > While the question of “Why Does Q-Greedy Policy Struggle with Optimal Datasets?” is one of the main questions that the submission focuses on that takes up more than two pages in the main body of the paper, but in reality this question has been only investigated in one game in the MuJoCo environment presented in the Appendix. I do not think this is sufficient enough evidence that the experiments of the submission support the claim that is being made here.
> >
> > Looking at the results reported in Appendix E, MuJoCo Hopper results, it seems that as the dimension of the action space increases the results reported in the main paper about the toy example becomes less relevant.
> >
> > I thank the authors for their rebuttal.  The submission needs substantial re-writing. I will keep my original score.

---

> ### Author Response · Authors · 2024-08-13
>
> We appreciate the reviewer's additional comments aimed at further clarifying our work.
>
> &nbsp;
>
> ### **POR changed their official score after their camera-ready version.**
>
> In the POR GitHub, around Nov 2022, questions were raised regarding the reproduction of POR results. Since the POR authors could not access their previous code due to the copyright issue, they re-implemented the work and presented the results anew. While most of the results matched the previous ones, they did not match the hopper-medium-v2 dataset. As a result, they replaced the score with the new results and updated the arxiv version accordingly. Through a comment on GitHub, the author recommended using the new results instead of the conference version, which the reviewer had checked. The exact author's comment is **"We have updated our paper in the arxiv to reflect that and the mismatch results in the hopper-medium-v2 dataset (lower it from 98.2 to 78.6). If you want to compare POR with your work, you can refer to this paper version."**. Due to NeurIPS rebuttal policies, we cannot include external links, but we provide the address of this comment: github.com/ryanxhr/POR/issues/2. Therefore, our POR score reporting is accurate and aligns with the author's intent.
>
> &nbsp;
>
> ### **Comparison with QT in Table R9.**
>
> Surpassing QT in 13 out of the 15 datasets we compared, with the remaining 2 scores being similar to QT within the range of standard deviation, cannot be a reason for rejection but rather demonstrates the superiority of QCS. (QCS mean 86.3 > QT mean 60.9)
>
> We want to emphasize that the offline RL benchmark features a variety of state and action dimensions, as well as different goals, making it a diverse and challenging benchmark where it is difficult for a single algorithm to perform well across all datasets compared to all baselines. For example, both POR and QT have instances where their performance is similar to or even lower than that of previous studies when examined closely at each dataset. For example, POR's performance of 76.6 on walker2d-medium-replay is lower than TD3+BC's 81.8, and QT's reported performance of 59.3 on antmaze-medium-diverse is lower than IQL's 70.0. However, POR and QT are recognized for their sufficiently good results due to their generally strong performance. **Slightly lower or similar scores to baselines on a few datasets cannot be grounds for rejection. We believe that what truly matters is how robustly the performance has improved compared to previous works.**
>
> &nbsp;
>
> ### **Comparison of QCS to the Best Score**
>
> As mentioned in the previous response, we were unable to confirm the exact evaluation metric for POR. Since we have already compared the scores from the POR paper provided by the POR author in Tables 2 and 3 of our original manuscript, we conduct an additional comparison for QT with the best score. Table R10 shows the results of the comparison using the best score. As can be seen from the table, the QCS score has generally increased compared to the running average score that we previously reported. Compared to QT, it also shows superior results on average in MuJoCo (six datasets are better, two are similar, and one is worse), and especially demonstrates better performance with a large gap in Antmaze. While we knew that reporting the best score would naturally result in better performance, we want to emphasize again, as mentioned in our previous response, that we believe the last running average score is a more appropriate metric for offline settings, which is why we reported using this metric. We have demonstrated better performance than QT, but we would also like to clarify once again that QT should not be a reason for rejection, as it qualifies as concurrent work under NeurIPS policies.
>
> &nbsp;
>
> **Table R10. Performance comparison between QCS (ours) and QT. We evaluate each algorithm based on the best score.**
>
> | Dataset            | QT (best)           | QCS (best)         |
> |--------------------|---------------------|--------------------|
> | halfcheetah-m      | 51.4±0.4            | **60.6**±0.3       |
> | hopper-m         | 96.9±3.1          | **99.7**±0.3       |
> | walker2d-m       | 88.8±0.5           | **92.3**±0.4       |
> | halfcheetah-m-r    | 48.9±0.3        | **55.5**±0.5       |
> | hopper-m-r       | 102.0±0.2           | **103.4**±0.5      |
> | walker2d-m-r     | **98.5**±1.1        | **98.6**±0.3       |
> | halfcheetah-m-e  | **96.1**±0.2        | 95.6±0.2           |
> | hopper-m-e     | **113.4**±0.4       | **113.1**±0.6      |
> | walker2d-m-e     | 112.6±0.6           | **118.3**±1.2      |
> | **average**      | **89.8**            | **93.0**           |
> | antmaze-u        | 96.7±4.7            | **100.0**±0.0      |
> | antmaze-u-d        | 96.7±4.7            | **98.0**±4.0       |
> | antmaze-m-d        | 59.3±0.9            | **100.0**±0.0      |
> | antmaze-l-d        | 53.3±4.7            | **92.0**±7.5       |
> | **average**        | 76.5                | **97.5**           |

---

> ### Author Response · Authors · 2024-08-13
>
> ### **Observation of Q Overgeneralization is one of our contributions.**
>
> Starting from the 164th sentence in our manuscript (slightly below the sentence mentioned by the reviewer)—"We present a simple experiment to verify that learning $Q_\theta$ indeed induces over-generalization when trained on optimal trajectories."—this phenomenon is illustrated throughout Fig. 4 and Fig. 5 in Section 3.2.
>
> &nbsp;
>
> ### **The topic 'Why Does the Q-Greedy Policy Struggle with Optimal Datasets?' is addressed in Section 3.2 by demonstrating the phenomenon of Q generalization and validating it through various methods across different domains.**
> Section 3.2 explains that the Q function tends to over-generalize when trained on an optimal dataset, which is why the Q-Greedy Policy struggles with optimal datasets. Throughout the section, we demonstrate that this overgeneralization phenomenon can particularly occur when training with an optimal dataset, using various domains including toy environment, Gym Inverted Double Pendulum, and MuJoCo Hopper, as well as various methods such as Q-value analysis and NTK analysis  (please see Figures 4 and 5 in our manuscript). The appendix provides an in-depth analysis that couldn’t be fully covered in the main text due to space constraints, but it is already addressed in the main body.
>
> &nbsp;
>
> ### **The results reported in Appendix E (the MuJoCo Hopper results) are additional analyses that connect to the Hopper results shown in Figure 5 of the main paper.**
>
> In Section 3.2, Figure 4, we present the results of a toy experiment, and immediately afterward, in Figure 5, we extend this experiment to demonstrate that a similar phenomenon occurs in both the Gym Inverted Double Pendulum and MuJoCo Hopper environments. We would like to note that the main paper includes not only the results of the toy experiment but also the results for Hopper. Therefore, the results in Appendix E are not a sudden comparison involving a change of domain and an expansion of the dimension from the toy example, but rather a more detailed analysis of the results presented for Hopper in Figure 5.

---

### Author Rebuttal · Authors · 2024-08-06

We express our deepest gratitude to all the reviewers for their time and effort in evaluating our work and providing valuable advice. Our responses to the reviewers' comments have been left as replies to each review. Moreover, due to the rebuttal word limit, we have posted the responses that could not be addressed to individual reviewers here and specified which reviewer each response is for. Additionally, the content regarding the standard deviation pointed out by the reviewer WnjH can be found in the PDF attachment.

&nbsp;

### **G1. Regarding Antmaze-medium-replay dataset. - reply to reviewer WnjH**

According to [Ref.1], there are six datasets (antmaze-umaze, antmaze-umaze-diverse, antmaze-medium-play, antmaze-medium-diverse, antmaze-large-play, antmaze-large-diverse) for the Antmaze task, and we could not find an antmaze-medium-replay dataset. Neither the baselines we compared nor the HIQL [Ref.2] mentioned by the reviewer referred to the antmaze-medium-replay dataset. However, we recognize that new benchmarks are continuously being updated and our knowledge may be incomplete. If the reviewer can provide information on where we can find the antmaze-medium-replay dataset or any prior work that evaluates using that dataset, we will conduct additional tests.

&nbsp;

[Ref.1] Fu, Justin, et al. "D4rl: Datasets for deep data-driven reinforcement learning." arXiv 2020.

[Ref.2] Park, Seohong, et al. "Hiql: Offline goal-conditioned rl with latent states as actions." NeurIPS 2023.

&nbsp;

### **G2. Regarding Section 3.2.  - reply to reviewer WnjH**

We'd like to kindly ask the reviewer for specific parts in section 3.2 that seem to be unnecessary. We would be happy to revise and improve our paper accordingly.

In Section 3.2, what we are trying to convey is not 'how the Q-learning update works,' but rather **analyzing why a Q function trained with an optimal dataset tends to over-generalize, and why this is not the case when trained with a medium-quality dataset.** This analysis is novel and has not been addressed in previous works. Since we are comparing and analyzing the Q function based on dataset quality, the analysis naturally begins with how the Q function is updated. However, this is merely the starting point of the analysis; the crucial point is how in-sample Q-learning progresses according to dataset quality. We believe that properly analyzing offline Q-learning in various settings helps establish a logical structure, provides insight into the problem definition, and naturally leads to the development of the QCS algorithm. We will update the manuscript to clearly convey our key points.

&nbsp;

### **G3. Regarding Standard deviation - reply to reviewer WnjH**

Thank you for pointing this out and allowing us to make our score reporting more complete. The standard deviations for Tables 1, 2, and 8 in our manuscript have been added to Tables R6, R7, and R8 in the PDF attachment within the global response. Since some baselines do not report the standard deviation for their own algorithms [Ref.3, Ref.4, etc.], we add standard deviation except for those works.

&nbsp;

[Ref.3] Kostrikov, Ilya, Ashvin Nair, and Sergey Levine. "Offline Reinforcement Learning with Implicit Q-Learning." ICLR 2022.

[Ref.4] Emmons, Scott, et al. "RvS: What is Essential for Offline RL via Supervised Learning?." ICLR 2022.

&nbsp;

### **G4. Regarding wrong bolding. - reply to reviewer WnjH**
Thank you for pointing out. We made a mistake with the bold in the hopper-medium-expert dataset and will correct it. We also noticed that for the hopper-medium-replay dataset, the correct SQL score is 99.7, but we mistakenly reported it as 101.7 higher than 99.7. Since 101.7 was the maximum score we previously marked in bold, we will correct the score and remove the boldface.

&nbsp;

### **G5. Impact of the context length K. - reply to reviewer fTMz**

We conducted additional experiments by varying the context length of QCS. As seen in the results, the QCS is highly robust to changes in context length. The QCS is based on the DC [Ref.5], which emphasizes the local context in reinforcement learning, allowing it to achieve good results even with a short context length.

&nbsp;

**[Table R1. QCS with varying context length $K$.]**

| Dataset | K=2 | K=8 | K=20 |
|---------|-----|-----|------|
| halfcheetah-medium | 60.2±0.6 | 59.0±0.4 | 59.2±0.3 |
| hopper-medium | 97.7±2.5 | 96.4±3.7 | 95.5±5.1 |
| walker2d-medium | 88.4±0.9 | 88.2±1.1 | 87.8±2.0 |
| halfcheetah-medium-replay | 54.0±0.4 | 54.1±0.8 | 52.6±0.8 |
| hopper-medium-replay | 100.1±1.5 | 100.4±1.1 | 99.0±3.4 |
| walker2d-medium-replay | 86.4±5.1 | 94.1±2.0 | 88.6±4.1 |

&nbsp;

[Ref.5] Kim, Jeonghye, et al. "Decision ConvFormer: Local Filtering in MetaFormer is Sufficient for Decision Making." ICLR 2024.

---

### Decision · Program_Chairs · 2024-09-25

**Decision:**

Accept (poster)

**Comment:**

The submission presents an analysis of the failure modes of RCSL and value-based approaches in Offline RL. While this analysis may not be particularly novel or surprising, it effectively establishes a solid foundation for developing a new solution that mitigates these failures. The proposed algorithm, though simple, delivers compelling empirical results that validate the soundness of the approach. Based on these strengths, I recommend accepting this submission as a poster.